# CircGRIA1 shows an age-related increase in male *macaque* brain and regulates synaptic plasticity and synaptogenesis

Kaiyu Xu [1,2,10], Ying Zhang[1,2,10], Wandi Xiong[3,4,5,10], Zhongyu Zhang[3,4], Zhengbo Wang [1,6], Longbao Lv[1,7], Chao Liu[7], Zhengfei Hu[7], Yong-Tang Zheng [1,7,8], Lin Lu[3,4,5], Xin-Tian Hu[1,7,8] & Jiali Li [1,3,4,7,8,9 ✉]

Circular RNAs (circRNAs) are abundant in mammalian brain and some show age-dependent expression patterns. Here, we report that circGRIA1, a conserved circRNA isoform derived from the genomic loci of α-mino-3-hydroxy-5-methyl-4-isoxazole propionic acid (AMPA) receptor subunit Gria1, shows an age-related and male-specific increase in expression in the *rhesus macaque* prefrontal cortex and hippocampus. We show circGRIA1 is predominantly localized to the nucleus, and find an age-related increase in its association with the promoter region of *Gria1* gene, suggesting it has a regulatory role in *Gria1* transcription. In vitro and in vivo manipulation of circGRIA1 negatively regulates *Gria1* mRNA and protein levels. Knockdown of circGRIA1 results in an age-related improvement of synaptogenesis, and GluR1 activity-dependent synaptic plasticity in the hippocampal neurons in males. Our findings underscore the importance of circRNA regulation and offer an insight into the biology of brain aging.

[1] Key Laboratory of Animal Models and Human Disease Mechanisms of Chinese Academy of Sciences & Yunnan Province, Kunming Institute of Zoology, Chinese Academy of Sciences, Kunming, Yunnan, China. [2] Kunming College of Life Science, University of Chinese Academy of Sciences, Kunming, Yunnan, China. [3] National Institute on Drug Dependence, Peking University, Beijing, China. [4] PKU/McGovern Institute for Brain Research, Peking University, Beijing, China. [5] Peking-Tsinghua Center for Life Sciences, Beijing, China. [6] Yunnan Key Laboratory of Primate Biomedical Research, Institute of Primate Translational Medicine, Kunming University of Science and Technology, Kunming, Yunnan, China. [7] Kunming Primate Research Center of the Chinese Academy of Sciences, Kunming, Yunnan, China. [8] National Research Facility for Phenotypic and Genetic Analysis of Model Animals, Kunming Institute of Zoology, Chinese Academy of Sciences, Kunming, Yunnan, China. [9] CAS Center for Excellence in Animal Evolution and Genetics, Chinese Academy of Sciences, Kunming, Yunnan, China. [10] These authors contributed equally: Kaiyu Xu, Ying Zhang, Wandi Xiong. ✉email: jialili@bjmu.edu.cn

Brain aging is characterized by multiple changes in cell structure; molecular, physiological, and biophysical properties of its resident neurons[1–4]. At a system level, there degenerative changes are accompanied by a decreased synaptic plasticity and occasionally by neuronal loss[5,6]. While descriptive studies of brain aging are numerous, molecular markers that offer insight into the mechanisms underlying the biology of brain aging remain elusive and undefined[4,7,8].

Age-related alterations in neuronal function are partially resulted from changes in homeostatic synaptic plasticity and calcium homeostasis due to such as factors like decreased expression of neurotransmitter receptors and selective ion channel messenger RNAs (mRNAs)[9–12]. Aging indeed reduced expression of glutamate receptors and their associated second messengers. As these play an important role in regulating intracellular calcium they are critical for the maintenance of synaptic plasticity. NMDA (N-methyl-D-aspartate) and AMPA receptors are the predominant excitatory neurotransmitter receptors in the mammalian brain. Cotman et al.[13] noted a 45% loss in glutamate receptors that occur in aged rats when compared with young animals[11,13]. Yet studies such as these are silent on the mechanisms behind this age-related decrease.

CircRNAs are a novel class of transcripts that are synthesized by head-to-tail splicing of linear RNA molecules. Their existence has been known for many years, but the consequences of their complex tissue- and spatiotemporal-expression patterns are not yet fully understood[14–17]. One major role of circRNAs is to serve as miRNA "sponges" that buffer the action of miRNAs[15,18]. Slightly deviated from that, for instance, muscle blind gene-derived circMbl modulates host mRNA production by competing with it for splicing and other maturation factors[19]. In addition, circRNAs also regulate the intracellular transport of RNAs through their abilities to sequester RNA-binding proteins[20]. Adding to the potential regulatory complexity, studies have shown that circRNAs can be translated into peptides in a cap-independent way[21,22]. Taken together, circRNAs represent a heterogeneous and dynamic class of noncoding transcripts that potentially regulates brain function through a series of diverse mechanisms.

CircRNAs are abundantly produced in the brain and are naturally present inside neurons[23], where they potentially serve as a new and relatively unexplored regulatory network that may be active in brain development maturation and aging[23–25]. Indeed, our recent work has described dynamic changes in circRNA expression in rhesus macaque brain during aging, and indicated that the complicate correlation between circRNA and host mRNA expression may be involved in the biology of brain aging[26].

In this study, we focus specifically on the AMPA receptor gene-derived circGRIA1. Utilizing postmortem brain tissues of macaque together with in vivo and in vitro manipulation of circGRIA1 expression, we disclose an age-related and male-biased increase in circGRIA1 expression in the male macaque brain that likely explains the loss of synaptic dysfunction over the aging states.

## Results

### Age- and sex-related changes in circGRIA1 expression in the macaque brain.
Previously, using deep RNA profiling, we described a comprehensive map of changes in circRNA expression in rhesus macaque (macaca mulatta) brain during aging[26]. The study explored the variable age-related correlations between circRNA and host mRNA expression. We identified 11 age-related circRNAs and host mRNAs whose functions make them prime candidates for serving to regulate brain aging. Of these eleven circRNAs, nine are negatively correlated with the aging processing, while two are positively correlated[26]. In this study, we further analyzed one of these 11 circRNAs, circGRIA1. We found a total of 12 circRNA isoforms derived from the genomic locus of Gria1, including 10 circRNA isoforms detected by find_circ[18] and 11 detected by CIRI2[27] (Fig. 1a and Supplementary Data 1). These circRNAs included 2 single exons, 9 multiple exons, and 1 intronic circRNA. Among these, circGRIA1, an evolutionally conserved isoform formed by exon 4, 5, 6 of Gria1 gene, was highly expressed in 20-year-old male samples (Fig. 1b). Significantly, circGRIA1 expression was inversely correlated with its host mRNA (Gria1) expression (Fig. 1c). The most recent circAtlas database[28], identifies 67, 34, and 16 circRNA isoforms derived from the genomic loci of Gria1 gene from human, mouse, and macaque, respectively. We found macaque homologues of both human and mouse for circGRIA1, indicating its conservation in mammals (Supplementary Data 2). The homologous circRNAs were detected from brain and spinal cord tissues in circAtlas, with higher expression in spinal cord.

To verify the negative correlation between circGRIA1 and its host mRNA, we analyzed extracts of RNA from frozen postmortem brain samples of 10- and 20-year-old male macaques. We first validated their expression by qPCR using divergent primers (Fig. 1d, e). Next, we performed northern blot using probes against either the junction site (which exclusively recognizes Gria1 circular transcript), or a probe from an exon (which recognizes both Gria1 linear and circular transcripts). We examined circGRIA1 expression in hippocampus of 10 and 20 years old male and female macaques (Fig. 1f, g and Supplementary Fig. 1a, b). Consistently, increased circGRIA1 expression was found exclusively in the older samples of male macaques. Sanger sequencing of RT-PCR products was aligned with Gria1 genome and further validated the identity of circGRIA1 (Fig. 1h). Different subcellular distributions and characteristics of circRNAs could be responsible for different regulatory functions, along with differences in their length, GC content, alternative circularization, and parental gene function[29]. Thus cytoplasmic and nuclear RNAs extracted from postmortem frozen hippocampal tissues of 20-year-old male macaque were analyzed by RT-qPCR to verify the subcellular localization of circGRIA1. Notably, more than half of circGRIA1 was found in the nuclear fraction suggesting a potential function there (Supplementary Fig. 1c).

### CircGRIA1 negatively correlates to Gria1 mRNA expression.
Next, we decided to investigate whether circGRIA1 expression was associated with the biological process of brain aging. First, using BASEscope in situ hybridization (ISH) for detection of circRNA with the junction site probes, we verified the age-related increase of circGRIA1 expression in prefrontal cortex (PFC), hippocampal CA1, and dentate gyrus (DG) of 20-year-old male macaque (Fig. 2a, b). Using RNAscope ISH for detection of linear RNA with the specific probe sets against host Gria1 mRNA, we examined the levels of Gria1 expression, and found an age-related decrease in PFC and hippocampus of 20-year-old male macaque brain (Fig. 2c, d). Interestingly, in 20-year-old female PFC and hippocampus, where there was no detectable age-related increase in circGRIA1 expression, we nonetheless found an age-related decrease in Gria1 expression was found (Supplementary Fig. 2). Data from immunohistochemistry (IHC) and western blot analyses further validated that the GluR1 decreased in PFC and hippocampus of 20-year-old both sexes (Supplementary Fig. 3). This suggests a different molecular basis to the regulation of Gria1 expression and function in the brains of male and female macaques. The protein levels of GluR2 also showed an age-related decreased (Supplementary Fig. 3), yet no circRNA isoform derived from GluR2 gene, Gria2, could be identified in either sex.

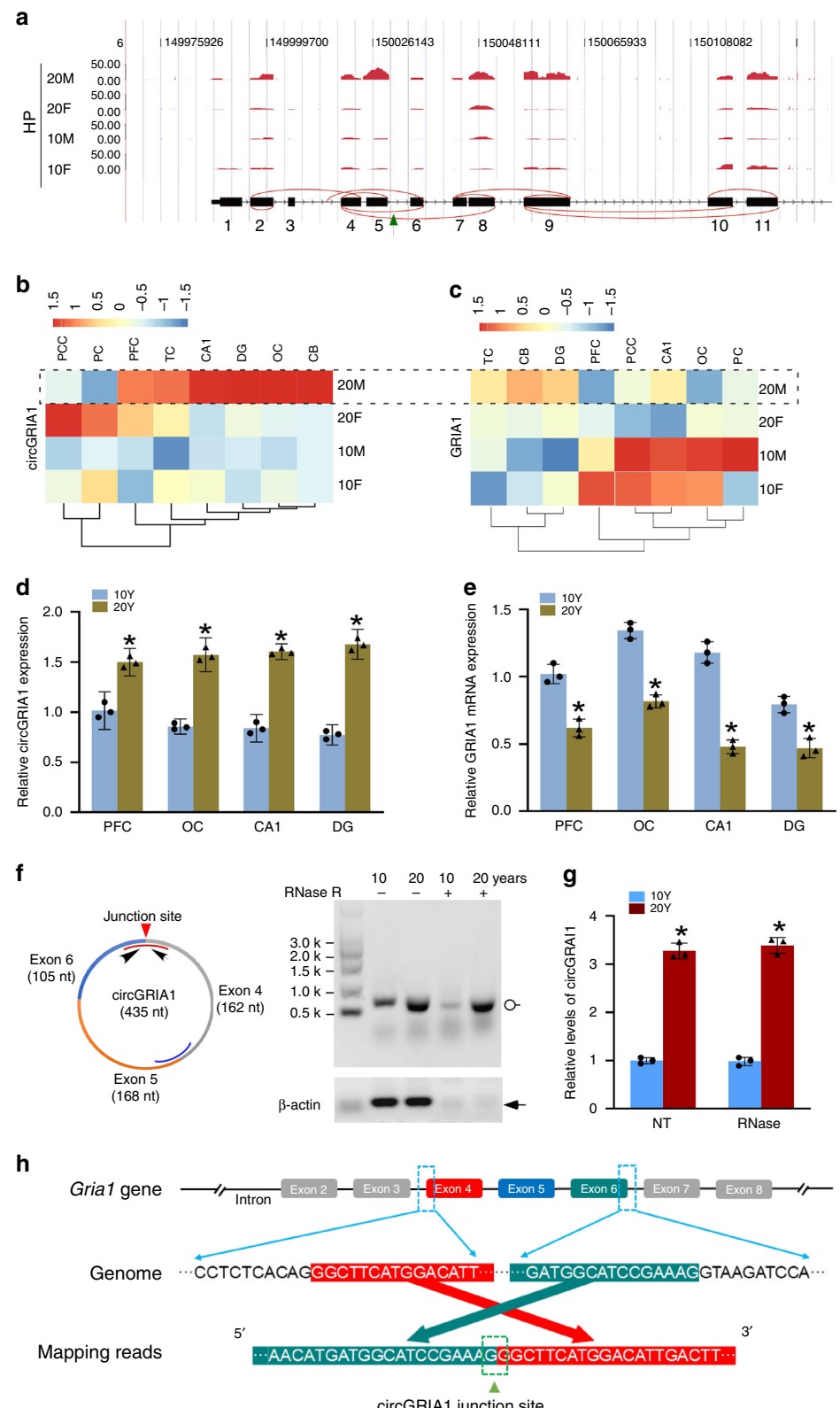

To investigate whether in vitro hippocampal cultures of fetal *macaque* might reproduce these in vivo age-related changes, we used BASEscope ISH and RT-qPCR to examine the levels of circGRIA1 expression in hippocampal cultures of male fetal *macaque* at 14 days in vitro (DIV14) and DIV28 (Supplementary Fig. 4). As with aging brain, we found increased levels of circGRIA1 expression in neurons at DIV28 compared with that of the DIV14. By contrast, data from RNAscope ISH and RT-qPCR showed decreases in *Gria1* expression in hippocampal cultures of male fetal *macaque* at DIV28 compared with that of the DIV14 (Supplementary Fig. 4).

**Fig. 1 Characteristics of circGRIA1 and *Gria1* mRNA expression. a** RNA-seq analysis reveals expression and maps of all circRNA isoforms derived from *macaque Gria1* gene in hippocampus of 10- and 20-year-old male (10 M, 20 M) and female (10 F, 20 F) *macaques*. Green arrow indicates the circGRIA1 investigated in the present study. **b** Heatmap of circGRIA1 expression showing the increased levels in several clusters at 20-year-old male *macaque* (20 M) compare with 10 years (10 M) ($p = 0.02$, $t$-test). **c** Heatmap of *Gria1* mRNA expression showing the decreased levels in several clusters at 20-year-old male (20 M) *macaque* compare with 10 years (10 M) ($p = 0.04$, $t$-test), which showed contrary pattern with circGRIA1 in (**b**). **d, e** qPCR validated the presence of circGRIA1 detected by CIRI2 and *Gria1* mRNA with the correlated male samples from PFC, OC, and hippocampal CA1 and DG for sequencing. Linear RNA of *Gapdh* was used as internal reference, and relative value of each sample was normalized by the first sample in each (*$p < 0.05$, unpaired $t$-test; mean ± S.D., $n = 3$ repetitions of the experiment of each sample from 2 to 3 animals per age group). **f** Representative northern blots of *Gria1* circular transcripts in postmortem frozen hippocampal tissues of 10- and 20-year-old (10 Y and 20 Y) male *macaques*. The blots are representative of replicates of three independent experiment. The curves represent circRNA junction-targeting (red) or -nontargeting (blue) sequences of biotin-labeled probes. $K$ denotes 1000 nt. **g** Relative intensities of northern blot signals illustrated in (**f**) were quantified by use of Image J. The linear and circular *Gria1* were quantified and calculated in comparison with actin (*$p < 0.05$, unpaired $t$-test; mean ± S.D.; $n = 3$ repetitions of the experiment of each sample from 2 to 3 animals per age group). **h** The sequence alignment of circGRIA1 and its host mRNA gene. A schematic representation of the exon structure is shown at the top. The gene is coded on the negative strand. The arrows indicate that the sanger sequencing data of semi-quantitative RT-PCR products of circGRIA1 are consistent with the coding sequences. Source data are provided as a Source Data file.

**Association of circGRIA1 with the promoter region of its parental gene**. Recent study reveals that the gene regulation functions of a circRNA are accomplished by its ability to compete with the factors needed to splice the linear native mRNA. In this way they serve as an important regulator of mRNA maturation[19,30]. To explore whether circGRIA1 negatively correlates to its host mRNA expression in the cis-acting manner, we investigated potential interactions between circGRIA1 and chromatin. Using chromatin isolation by RNA purification (ChIRP) we examined genomic DNA coprecipitated with circGRIA1 through biotin-labeled oligonucleotides complementary to circGRIA1 junction sites. We focused our analysis on the 5′-UTR and 3′-UTR regions of the *Gria1* gene elements that were coprecipitated with circGRIA1 from both in vivo and in vitro preparations. In our samples, ChIRP with circGRIA1 pulled down endogenous circGRIA1 itself, but failed to precipitate the normal, linear mRNA transcript. As negative controls we used 5.8S RNA and circGRIN2A (Fig. 3a–c). Importantly, when ChIRP with circGRIA1 was performed on lysates from hippocampal tissues of 20-year-old *macaques*, a substantial amount of PCR product complementary to the 5′-UTR of the *Gria1* gene was recovered, especially compared with comparable assay using brain samples from 10-year-old animal (Fig. 3d, e). No detectable 3-UTR genomic DNA was found. Sanger sequencing of RT-PCR products verified the identity of the *Gria1* 5′-UTR (Fig. 3f). Notably, overexpression circGRIA1 in hippocampal cultures of fetal *macaque* significantly increased its association with the 5′-UTR of the parental gene at DIV28 (Supplementary Fig. 5a, b).

To determine whether association circGRIA1 with the promoter region of its parental gene affects its transcription, SH-SY5Y cells, which express no detectable circGRIA1, were transfected with the 5′UTR sequence (~330 bp) of *macaque Gria1* in pGL4.11 vector together with either circGRIA1 in Tet-on circRNA vector or pLCDH-ciR vector. Twenty-four hours later, we assayed for luciferase activity. We found that the relative levels of luciferase activity were consistent with our hypothesis that circGRIA1 strongly downregulates transcription from the *Gria1* gene by its competitive association with the promoter region (Supplementary Fig. 5c, d). Next, using RNA–DNA dual fluorescent in situ hybridization (FISH) we further validated co-localization of nuclear circGRIA1 with the genomic loci of *Gria1* in hippocampal cultures of male fetal *macaque* at DIV28 compared with that of the DIV14 (Supplementary Fig. 6).

**CircGRIA1 negatively regulates *Gria1* mRNA expression**. Next, using BASEscope and RNAscope ISH we examined correlation between the levels of circGRIA1 and *Gria1* expression in hippocampal cultures of male fetal *macaque* at DIV14 and DIV28.

We introduced the junction site-targeting siRNAs against circGRIA1 as well as the mismatched junction site-targeting siRNA-control at DIV5. Analysis of the average intensities of BASEscope ISH signals revealed a significant reduction of circGRIA1 expression at DIV28 infected with viral particles of siRNA (Fig. 4a–c). By contrast, analysis of the average intensities of RNAscope ISH signals revealed that knockdown of circGRIA1 led to a significant increase in *Gria1* expression at DIV28 (Fig. 4a–c). Notably, no substantial response was seen at DIV14 with introduction of siRNA consistent with the barely detectable levels of endogenous circGRIA1. Since circRNAs are quite stable, we sought to exclude the effect of age-related accumulation of circGRIA on the negative correlation between circGRIA1 and its host mRNA expression, nascent RNA was purified by use of the Click-iT Nascent RNA Capture Kit followed by qPCR. Indeed, both knockdown and overexpression of circGRIA1 could reverse the changes in *Gria1* mRNA expression (Fig. 4c). Western blots of protein levels were consistent with the idea that manipulation of circGRIA1 expression led to age-related changes in *Gria1* mRNA expression as well as GluR1 protein (Fig. 4d, e).

**CircGRIA1 contributes to synaptogenesis**. Brain aging entails many chemical, biological, and structural changes including synaptogenesis[31–33]. NMDA and AMPA receptor activities are required for both long-term potentiation and neural activity-dependent synaptogenesis[34]. To investigate whether the negative correlation between circGRIA1 and *Gria1* expression is involved in regulation of pre- and post-synaptogenesis over the aging states, we first microinjected AAV viral particles of siRNAs against circGRIA1 into hippocampus of 10- and 20-year-old male and female *macaques* under the guidance of magnetic resonance imaging (MRI) (Supplementary Fig. 7). Six weeks later, the levels of circGRIA1 and *Gria1* expression were examined using BASEscope ISH (Fig. 5a, b and Supplementary Fig. 8a, b), RNAscope ISH (Fig. 5d, e and Supplementary Fig. 8d, e), and RT-qPCR (Fig. 5c, f and Supplementary Fig. 8c, f). Knockdown of circGRIA1 led to significant decreases in circGRIA1 expression, and increases in *Gria1* mRNA expression in the hippocampal neurons of male but not female *macaque*. Next, we examined age-related changes in the levels of several synaptic components in the brains of *rhesus macaque*s. IHC showed significant decreases in the levels of synapsin-I and PSD95 in PFC and hippocampus (CA1 and DG) of 20-year-old male and female *macaques* compared with that of the 10 years (Fig. 6a, c and Supplementary Fig. 9a, c). In addition, despite little change in VAMP2, both pre- and postsynaptic components including synapsin-I, synaptotagmin-I, syntaxin-2, PSD95, and neuroligin-1 all showed

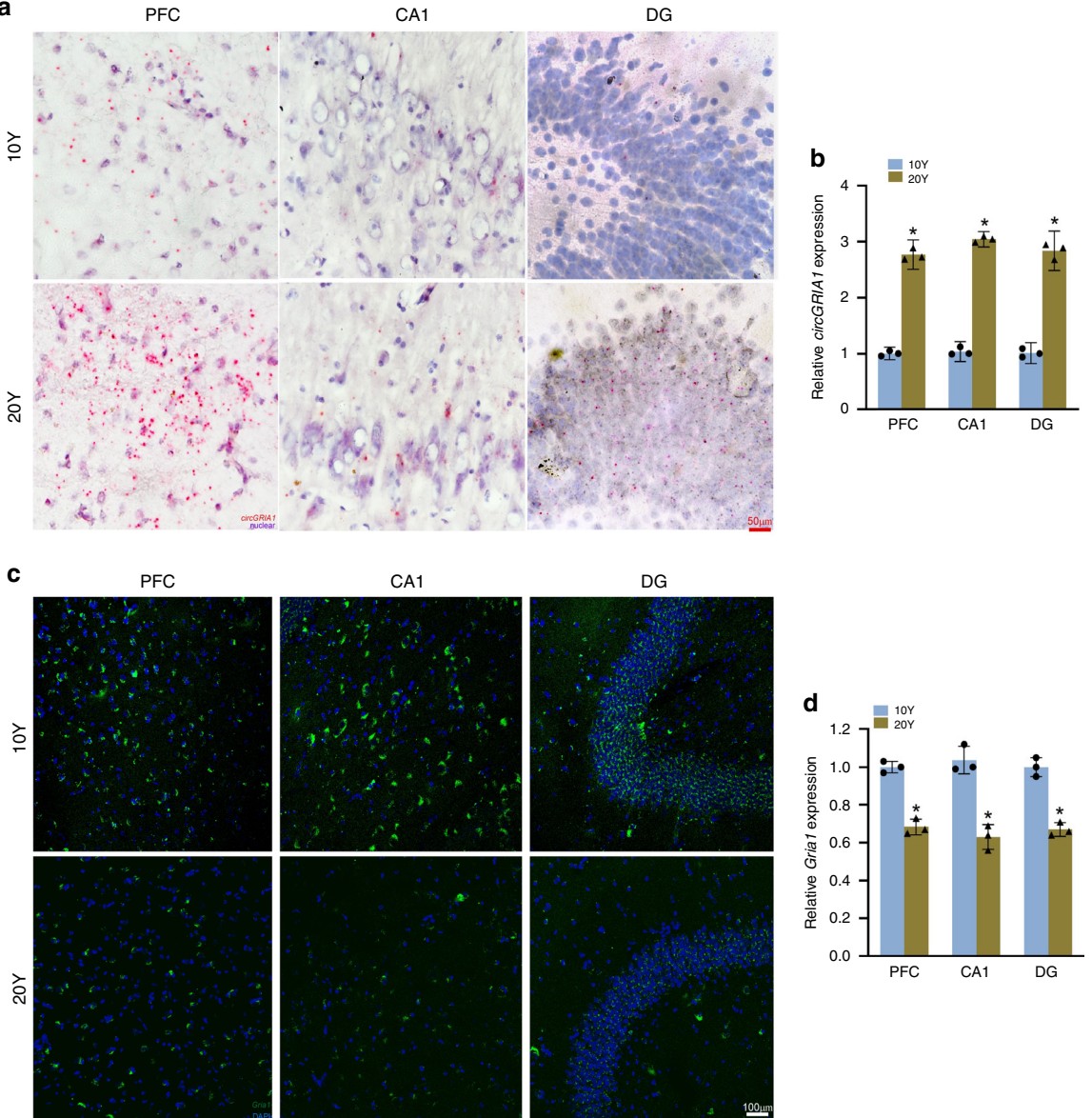

**Fig. 2 Negative correlation between CircGRIA1 and *Gria1* mRNA expression. a** Representative images of BASEscope in situ hybridization (ISH) showing circGRIA1 expression in PFC and hippocampus (CA1 and DG) of 10-Y and 20-Y male *macaques*. The images are representative of three independent experiments of each sample from 2 to 3 animals per age group. Red dots indicate circGRIA1 with nuclei counterstaining by hematoxylin. Scale bar, 50 μm. **b** Relative intensities of BASEscope ISH signals of circGRIA1 illustrated in (**a**) were quantified by use of Image J. Data are present as mean ± S.D. ($n = $ 45–60 cells per age group; *$p < 0.05$, unpaired *t*-test). **c** Representative images of RNAscope ISH showing *Gria1* expression in PFC and hippocampus of 10-Y and 20-Y male *macaques*. The images are representative of three independent experiment of each sample from 2 to 3 animals per age group. Green dots indicate *Gria1* mRNA with nuclei counterstaining by DAPI. Scale bar, 100 μm. **d** Relative intensities of RNAscope ISH signals of *Gria1* illustrated in (**c**) were quantified by use of Image J. Data are present as mean ± S.D. ($n = 50$–65 cells per age group; *$p < 0.05$, unpaired *t*-test). Source data are provided as a Source Data file.

age-related decreases in the brain tissues of 20-year-old male *macaques* (Fig. 6d, e and Supplementary Fig. 10a, b).

To determine whether knockdown of circGRIA1 prevents the decreases in the presynaptic vesicle protein, synapsin-I and the postsynaptic protein, PSD95, cryostat sections of postmortem hippocampus from 10- and 20-year-old male and female *macaques* in which circGRIA1 has been knocked down were examined. Knockdown of circGRIA1 robustly increased the levels of synapsin-I, in the hippocampal neurons of 20-year-old male *macaque* but not that of 20-year-old female *macaque* (Fig. 6b, f and Supplementary Fig. 9b, d). Curiously despite these findings, in vivo knockdown of circGRIA1 led to few changes in the levels

of PSD95 in either male and female *macaques* at 20 years of age. Indeed, similar changes were also found in hippocampal cultures of male fetal *macaque* with introduction of siRNA (Supplementary Fig. 10c–f). Despite significant circGRIA1-induced decreases in the densities of both of synapsin-I and PSD95 found at DIV28, the cultures infected with siRNAs at DIV5 showed significantly increases in the densities of synapsin-I at DIV28 with little effect on PSD95.

**CircGRIA1 regulates synaptic plasticity and Ca$^{2+}$ homeostasis.** Homeostatic synaptic plasticity is associated with AMPA and

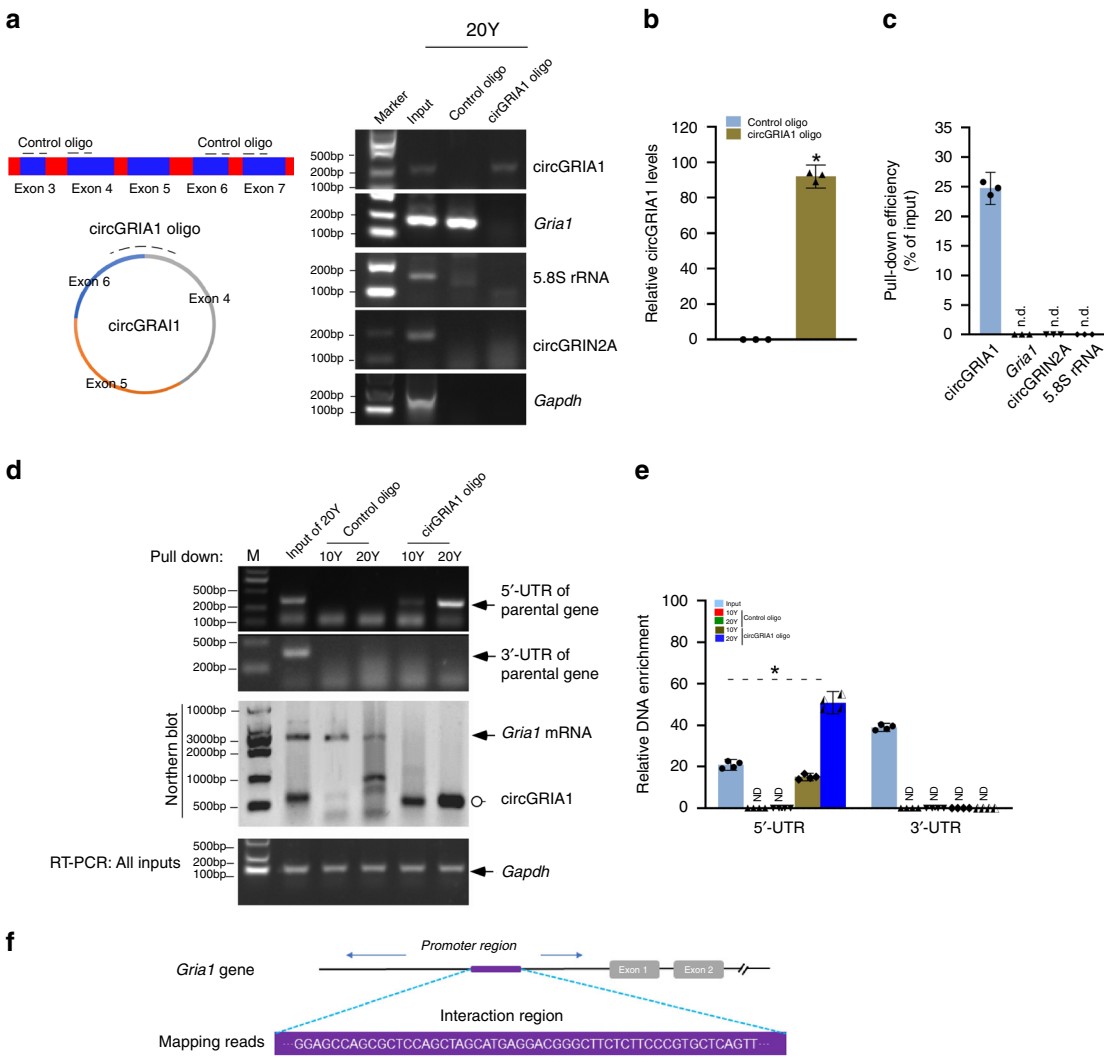

**Fig. 3 The regulatory effect of circGRIA1 on the parental genomic locus. a** Biotin-labeled oligonucleotides (oligos) complementary to the junction sequences of circGRIA1 were used for Chromatin Isolation by RNA purification (CHIRP). Oligos complementary to upstream and downstream exons were used as control. Total RNA was extracted from postmortem frozen hippocampal tissues of 20-Y male *macaque*. The images are representative of three independent experiments of each sample from 2 to 3 animals per age group. **b**, **c** Quantification of three repetitions of independent experiment in (**a**) showed the CHIRP efficiency of circGRIA1. Pre-mRNA of *Gria1* was not detected (n. d.) in the CHIRP products as examined with primer pairs corresponding to the parental gene. Both circGRIN2A and 5.8S rRNA were used as controls. Black dash curves represent circRNA junction-targeting (curve) and -nontargeting (linear) sequences of biotin-labeled oligos. **d** CircGRIA1 interacts with the promoter of its parental gene. Total RNA was extracted from postmortem frozen hippocampal tissues of 10- and 20-year-old male *macaques*. Upper panel shows PCR products of the parental gene 5′-UTR and 3′-UTR regions against circGRIA1 oligos in ChIRP. Middle panel shows northern blot of Gria1 mRNA and circGRIA1 in the product of CHIRP with digoxin-labeled probes. Semi-quantitative RT-PCR of *Gapdh* was a loading control. **e** Quantification of three repetitions of independent experiment in (**d**). Data represents mean ± S.D.; *$p < 0.01$, unpaired *t*-test. **f** The alignment of circGRIA1-associating motif and the promoter region of its host mRNA gene. A schematic representation of the promoter structure is shown at the top. The gene is coded on the negative strand. The arrows indicate the sanger sequencing data of semi-quantitative RT-PCR products of circGRIA1 parental gene 5′-UTR from CHIRP consistent with the coding sequences. Source data are provided as a Source Data file.

NMDA (*N*-methyl-D-aspartate) receptors-mediated neuronal activity[35]. Changes in the mini excitatory postsynaptic currents (mEPSCs) are used to model this activity as their levels are dependent on the number of synapses formed and/or the presynaptic rate of vesicle release. At equilibrium, mEPSCs amplitude is dependent on either postsynaptic glutamate responsiveness or a presynaptic glutamate content of synaptic vesicles or both. Since circGRIA1 not only affects GluR1 expression, but also regulates the densities of presynaptic component synapsin-I, we were curious whether circGRIA1 contributed to homeostatic synaptic plasticity. Hippocampal cultures of fetal *macaques* were prepared, then infected on DIV5 with the junction site-targeting siRNAs against circGRIA1 or the mismatched junction site-targeting siRNA-control. We then tracked the effects of this manipulation on the mEPSCs at different times after infection. Both the amplitude and frequency of spontaneous mEPSCs showed the normal decreases in control cultures (siRNA-control) at DIV28 neurons compared with DIV14. By contrast, siRNA against circGRIA1, when introduced into male neurons significantly attenuated this decreases (Fig. 7a–c). We next asked whether circGRIA1 also contributed to GABA_A receptor blockade-induced change by manipulating neuronal activity with bicuculline (a GABA_A receptor antagonist) treatment. At DIV28, bicuculline enhanced excitatory neural activity,

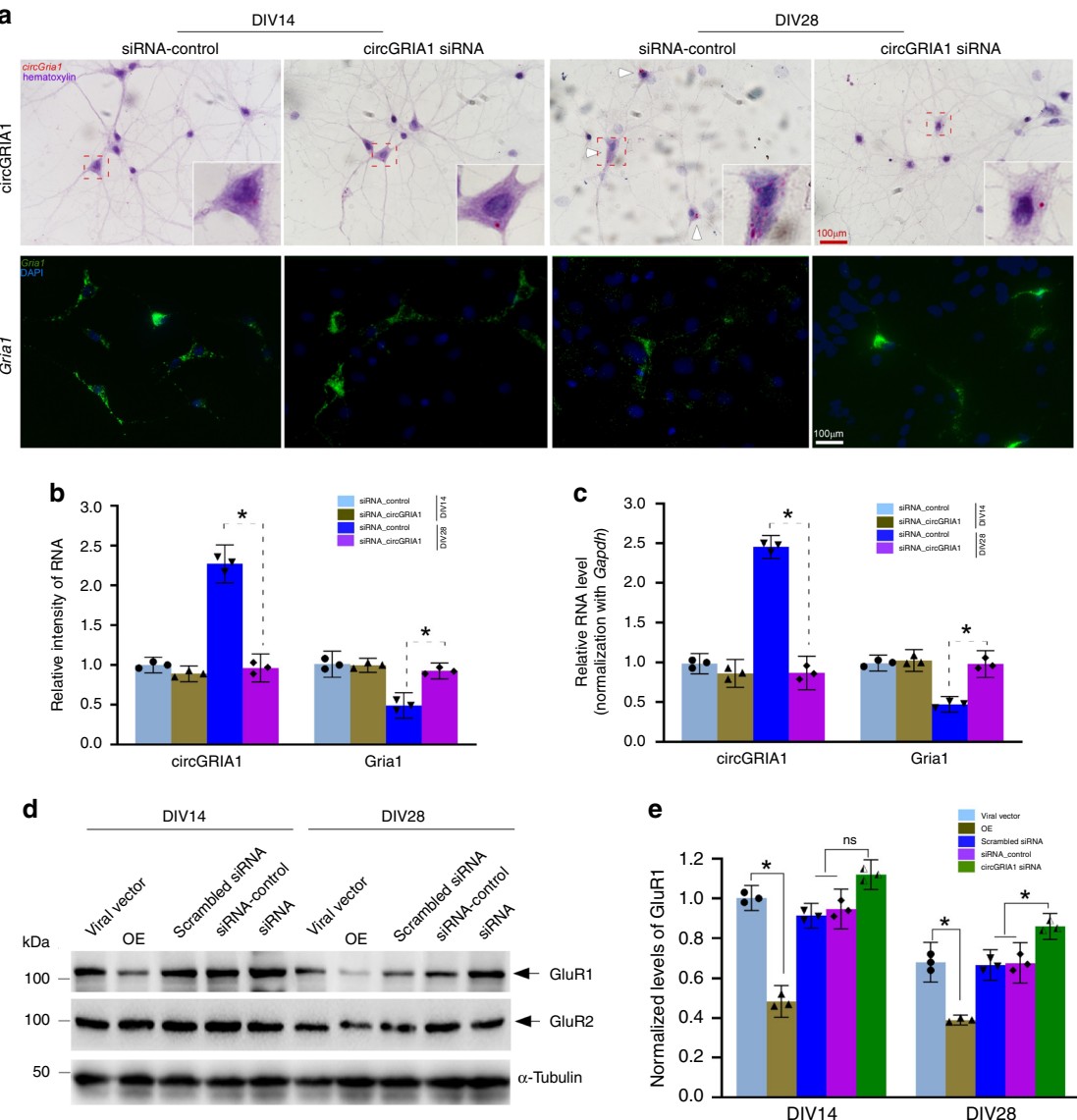

**Fig. 4 CircGRIA1 negatively regulates *Gria1* mRNA expression. a** Representative images of BASEscope and RNAscope ISH showing circGRIA1 and *Gria1* expression in hippocampal cultures of male fetal *macaque* at DIV14 and DIV28. The images are representative of three independent experiment of each sample from 2 to 3 fetal cultures per group. Red boxes indicate areas magnified. Scale bar, 100 μm. **b** Average intensities of circGRIA1 and *Gria1* signals per neuron illustrated in (**a**) were quantified by use of Image J. Data are present as mean ± S.D. ($n = 30–39$ per group; *$p < 0.01$), $p$ values were calculated using two-tailed Student's $t$ test or one-way ANOVA with Sidak's correction for multiple comparisons. **c** Negative correlation between nascent circGRIA1 and *Gria1* expression. Data are present as mean ± S.D. Each bar represents the average of three independent experiment; error bars denote S.D.; *$p < 0.05$, $p$ values were calculated using two-tailed Student's $t$ test or one-way ANOVA with Sidak's correction for multiple comparisons. **d** Protein extracts from DIV14, DIV28 male fetal *macaque* hippocampal neurons infected at DIV5 with viral circGRIA1, or siRNAs against circGRIA1 or control, were immunoblotted with GluR1 antibody. α-Tubulin was loading control. The blots represent three independent experiment of each hippocampal culture from 2 to 3 male fetal *macaques*. **e** Relative intensities of immunobloted signals of GluR1 illustrated in (**d**) were quantified by use of Image J. Data are present as mean ± S.D. Each bar represents the average of three independent experiment; error bars denote S.D.; n.s. no significance; *$p < 0.05$, $p$ values were calculated using two-tailed Student's $t$ test or one-way ANOVA with Sidak's correction or two-way ANOVA with Turkey's correction for multiple comparisons. Source data are provided as a Source Data file.

leading to a homeostatic decrease in the mEPSCs amplitude but not frequency. Interestingly, similar to the observed patterns of spontaneous mEPSCs, the changes in mEPSC amplitude but not frequency induced by bicuculline were unaffected by reducing circGRIA1 expression (Fig. 7a–c). To further confirm this finding, we induced chemical LTP (cLTP) at DIV14 and DIV28, and found that circGRIA1 levels in male neurons affect cLTP as well as mEPSCs (Fig. 7d).

Age-related neural plasticity deficits are tightly associated with age-induced alterations in calcium homeostasis[36]. This has

important consequences for the nerve system as both AMPA and NMDA receptors require calcium influx for their functioning[35,37]. A large body of evidence suggests that glutamate receptor signaling-induced disturbances of calcium homeostasis may be a causative factor in the biology of brain aging as well as Alzheimer's disease[7,9]. We therefore sought to determine whether increased levels of circGRIA1 expression could have a potential role in disturbances of AMPA receptor activity-dependent calcium homeostasis. We used a microscope-based fluorimeter to examine enriched populations of Fura-2-loaded female and

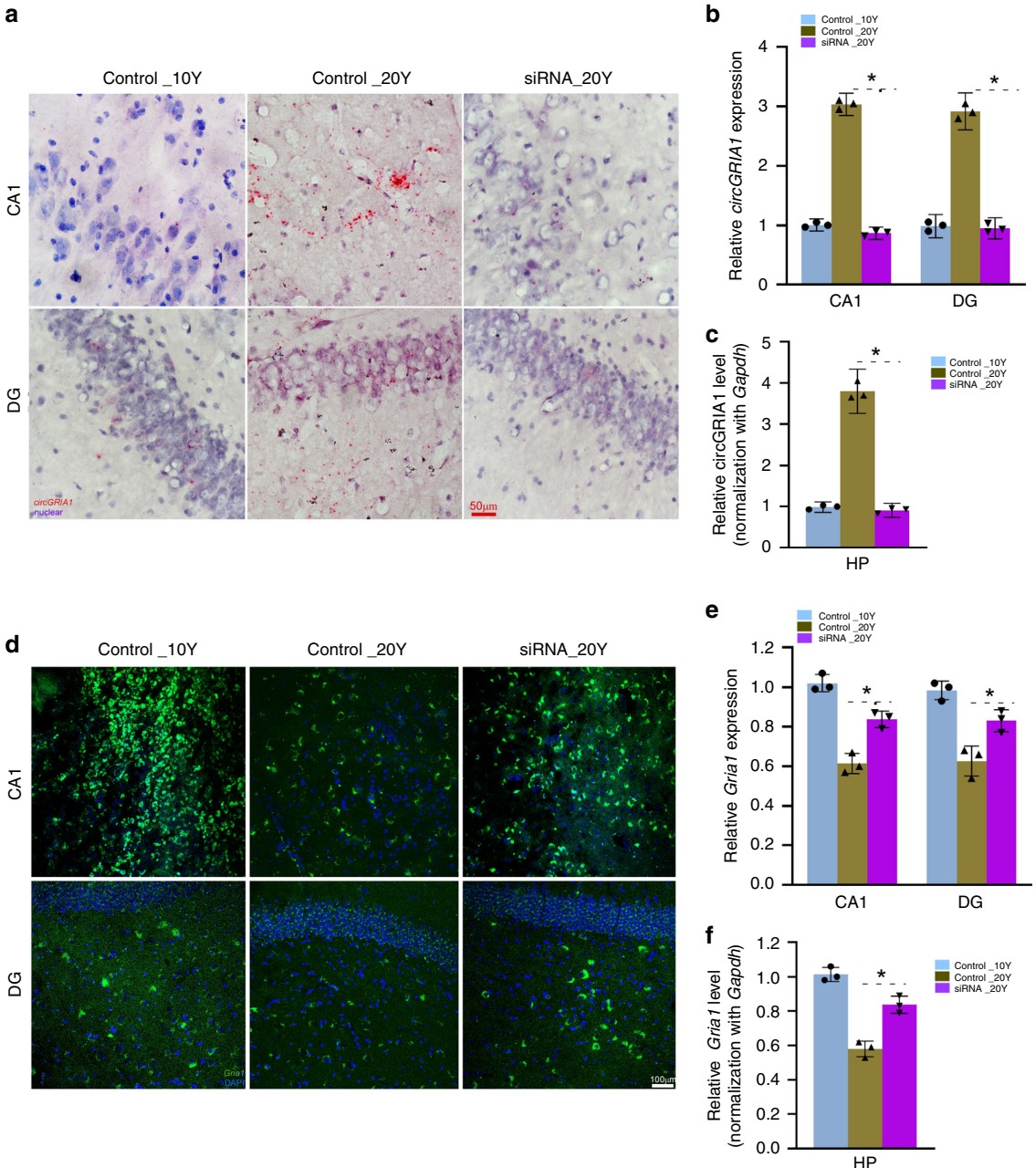

**Fig. 5 Knockdown of circGRIA1 increases *Gria1* expression. a** BASEscope ISH showing circGRIA1 expression in macaque hippocampus s. Ten-micron cryostat sections were performed BASEscope ISH (red dots with nuclei counterstaining by hematoxylin). Scale bar, 50 μm. The images are representative of three independent experiments of each sample from 2 to 3 animals per age group. **b** Relative intensities of BASEscope ISH signals of circGRIA1 illustrated in (**a**) were quantified by use of Image J. Data are present as mean ± S.D. ($n = 35$–45 cells per group; *$p < 0.05$, unpaired $t$-test). **c** RT-qPCR was performed for validation of circGRIA1 expression. Data are presented as mean ± S.D.; $n =$ three independent experiment of each sample from 2 to 3 animals per age group; *$p < 0.05$, unpaired $t$-test. **d** RNAscope ISH showing *Gria1* mRNA expression. Ten-micron cryostat sections were performed RNAscope ISH (green dots with nuclei counterstaining by DAPI). Scale bar, 100 μm. The images are representative of three independent experiment of each sample from 2 to 3 animals per age group. **e** Relative intensities of RNAscope ISH signals of *Gria1* illustrated in (**d**) were quantified by use of Image J. Data are present as mean ± S.D. ($n = 60$–75 cells per group; *$p < 0.05$, unpaired $t$-test). **f** Total RNAs were extracted from postmortem frozen hippocampal tissues of 10- and 20-year-old female *macaques* with microinjection of viral particles containing siRNAs against circGRIA1 or siRNA-control into hippocampus of *macaque* brains as indicated. qPCR was performed for validation of *Gria1* expression. Data are presented as mean ± S.D.; $n =$ three independent experiments of each sample from 2 to 3 animals per age group; *$p < 0.05$, unpaired $t$-test, error bars denote S.D. Source data are provided as a Source Data file.

male fetal *macaque* hippocampal neurons. We tracked Fura-2 fluorescence in groups of six to eight neurons after stimulation with glutamate followed by 100 μM cyclothiazide (CTZ, an AMPA receptor modulator, which binds to and desensitizes the AMPA receptor) treatment. The intracellular neuronal calcium concentrations, normally 38.6 ± 12.5 nM, increased and maintained at 150.9 ± 33.8 nM (~1.5-fold from the original baseline) after stimulation with Glutamate followed by CTZ treatment at DIV28 neurons (Fig. 7e). On the contrary, circGRIA1 knockdown in hippocampal cultures of male but female fetal *macaques* led to

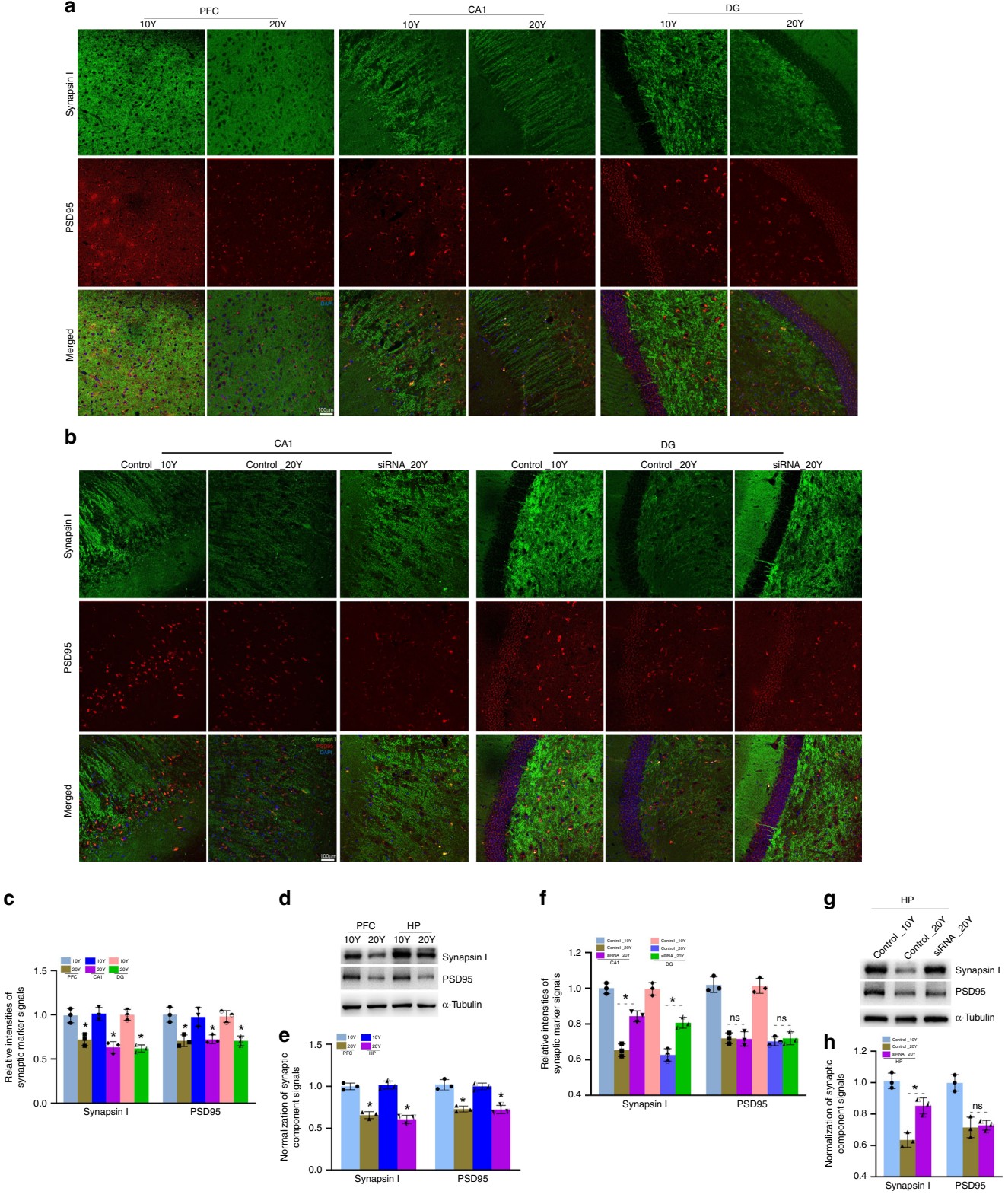

a significant increase in the intracellular calcium concentrations after stimulation with Glutamate followed by CTZ treatment at DIV28, which is nearly recovered to its level in DIV14 neurons. The intracellular calcium concentrations were calculated to range from 49.8 ± 12.5 nM and increased to and maintained around 206.8 ± 28.6 (~4.1-fold from the original baseline) after stimulation with Glutamate followed by CTZ treatment at DIV28 neurons.

## Discussion

During the normal course of eukaryotic transcript, the original RNA transcript is a precursor mRNA that is spliced to remove noncoding regions and form a mature mRNA. With circRNA this process is subverted and one or more noncanonical splicing events occurs {Salzman et al.[38] #72}{Zhang et al.[39] #78}. We have previously reported that there are a huge number of brain-specific circRNAs, many of which show age-related changes in their

**Fig. 6 Knockdown of circGRIA1 increases synaptogenesis. a** Representative images of immunofluorescence (IF) showing levels of synapsin-I and PSD95 in hippocampal (CA1 and DG) of 10-Y and 20-Y male *macaques*. Ten-micron cryostat sections were immunostained with synapsin-I (green) and PSD95 (red) antibodies. Scale bar, 100 μm. **b** Representative images of IF showing levels of synapsin-I and PSD95 in hippocampus of 10-Y and 20-Y male *macaques* after in vivo knockdown of circGRIA1. Hippocampal ten-micron cryostat sections were immunostained with synapsin-I (green) and PSD95 (red) antibodies. Scale bar, 100 μm. **c** Relative intensities of synapsin-I and PSD95 immunostaining illustrated in (**a**) were quantified by use of Image J. Data are present as mean ± S.D. (*n* = 2–3 animals per age group). Each bar represents the average of three independent experiment; error bars denote S.D. (*$p <$ 0.05, unpaired *t*-test). **d** Protein extracts from postmortem frozen PFC and hippocampal tissues of 10-Y and 20-Y male *macaques* were immunoblotted with synapsin-I and PSD95 antibodies. α-Tubulin was loading control. The blots represent three independent experiments of each sample from 2 to 3 animals per age group. **e** Relative intensities of immunoblotted signals of synapsin-I and PSD95 illustrated in (**d**) were quantified by use of Image J. Data are present as mean ± S.D.; *$p < 0.05$, unpaired *t*-test. **f** Relative intensities of synapsin-I and PSD95 immunostaining illustrated in (**b**) were quantified by use of Image J. Data are present as mean ± S.D. (*n* = 2–3 animals per age group) (*$p < 0.05$, unpaired *t*-test). **g** Protein extracts from frozen hippocampal tissues of 10-Y and 20-Y male *macaque* were immunoblotted with synapsin-I and PSD95 antibodies. α-Tubulin was loading control. The blots represent three independent experiment of each sample from 2 to 3 animals per age group. **h** Relative intensities of immunoblotted signals of synapsin-I and PSD95 illustrated in (**g**) were quantified by use of Image J. Data are present as mean ± S.D.; *$p < 0.05$, unpaired *t*-test, error bars denote S.D. Source data are provided as a Source Data file.

expression as compared to their linear mRNA during brain aging[26]. While the molecular mechanism and physiological function of changes in circRNA expression in the biology of brain aging remains unclear, we report here an example of the molecular consequences of such a change. Based on our data from circGRIA1, we propose that age-related increases in circRNA expression are the result of increased noncanonical splicing of newly generated transcripts in postmitotic neurons, and play an important role in the biology of brain aging.

CircGRIA1, a conserved but previously unannotated circRNA isoform derived from AMPA receptor GluR1 locus, increases in the brain of *rhesus macaque* in an age- and gender-related fashion. The increase has a substantial impact on glutamate receptor levels and therefore on synaptic plasticity. Combining in vivo and in vitro systems, we have investigated the regulatory role of circGRIA1 in the aging process of brain. We found that circGRIA1 expression is negatively associated with its host gene *Gria1* mRNA expression. Unexpectedly, the most robust correlation between circGRIA1 and its host mRNA gene expression in both *macaque* brain and fetal hippocampal cultures occurs in males. The effect is cell autonomous as well find it to be true both in whole brains as well as in dissociated cells in culture.

Our finding that circGRIA1 negatively regulates the transcriptional activity of its host *Gria1* gene via its association with the promoter region of the parental genomic locus offers potential insight into the mechanism by which gene expression is tuned by circRNA-chromatin association. We note that circGRIA1 theoretically has no direct DNA-binding ability. This suggests that its increased association with its parental gene 5′-UTR may involve other regulatory binding partners. This suggestion receives support from our finding of a strong gender bias to the action of circGRIA1 coupled with previous studies showing that sex-biased expression of protein-coding genes in the brain can be the result of multiple factor such as hormones and other epigenetic regulations[40,41]. Thus of nuclear receptors for sex hormones may be assisting in the sex-biased circRNA expression we observed in macaque brain. This should be a fertile avenue for future exploration since the majority of circRNAs have low expression in the brain, and only a few have been shown to serve as microRNA sponges[42].

Voltage-gated glutamate receptors such as NMDAR and AMPAR are critical to stable neural network function[43]. GluR1, one of the glutamate receptor subunits, is a part of postsynaptic and predominant excitatory ligand gated ion channel in the mammalian brain. It has been well documented that NMDA/ AMPA receptor-dependent neural activity is required for both pre- and post-synaptogenesis during neurodevelopment[37]. Despite its strong presynaptic effect on synaptogenesis, synapsin-

I levels and mEPSCs, changes in circGRIA1 expression have little impact on the postsynaptic marker PSD95. PSD95 is a stable scaffold protein that plays an important role in directing AMPA receptor membrane anchoring in the postsynaptic membrane[44], but there is no report of PSD95 being regulated by AMPA receptor activity. We propose that the circGRIA1-dependent changes in glutamate receptor levels may have trans-acting effect on presynaptic axon terminal as revealed by changes in synapsin-I expression, yet have little effect on the stability of postsynaptic structure itself, including its scaffolding proteins. It will be of interest to address whether and how circGRIA1 is involved in the regulation of the expression of other synaptic component genes during the aging process.

In the aggregate, our assays point to increased circGRIA1 expression in the aged brain making great contribution to the deregulation of synaptic plasticity through its regulatory effect on its host *Gria1* mRNA expression. Validation of this contribution comes from several independent lines of evidence involving in vivo microinjection of viral particles of siRNA against circGRIA1 into the hippocampus of *macaque*, and in vitro hippocampal cultures of fetal *macaque* neurons. We have assembled a quantitative and qualitative picture of the biological function of circGRIA1. We exploded three distinct aspects of the interplay of circGRIA1, synaptic plasticity, and calcium homeostasis. We also show the evolution of these effects with age in the nonhuman primate brains, independent of genetic influences. We believe that our in depth characterization of these molecular events will enhance future explorations of circRNA-mediated regulation, and function underlying the biology of brain aging with its associated age-related mental disorders.

## Methods

**Antibodies and chemical regents**. Antibodies against GluR1 (ab31232), GluR2 (ab133477), GFP (ab1213, ab183734), PSD95 (ab2723, ab13552), synapsin-I (ab64581), syntaxin-2 (ab233275), synaptotagmin (ab13259), VAMP2 (ab181869), neuroligin-1 (ab153821), α-tubulin (ab7291), and MAP2 (ab5392) were purchased from Abcam. Secondary antibodies used for immunocytochemistry were: Alexa 488-labeled chicken anti-mouse or anti-rabbit; Alexa 594-labeleddonkey anti-mouse or anti-rabbit (Invitrogen, Eugene, OR); all used at a dilution of 1:500. 4′,6-Diamidino-2-phenylindole (DAPI) (4′, 6′-diamidino-2-phenylindol) (Invitrogen) was used as a nuclear counterstain at 1 μg/ml. CTZ was purchased from Tocris Bioscience, and Furo-2, AM (F1201) was ordered from ThermoFisher. Bicuculline (S2694) and 5-Fluoro-2′-deoxyuridine (FDU) (F3503) were purchased from Sigma Aldrich.

**Constructs and plasmids**. All plasmids were constructed with restriction-enzyme digestion and ligation or alternative with recombinant methods. Oligos for all plasmid construction, probe preparation, siRNAs, and biotin-oligonucleotides are listed in Supplementary Tables 2 and 3. The AAV-shRNA vectors against circGRIA1 were purchased from Vigene Biosciences with scrambled shRNA as negative control. Viral stocks were stored at −80 °C until use. All plasmids were

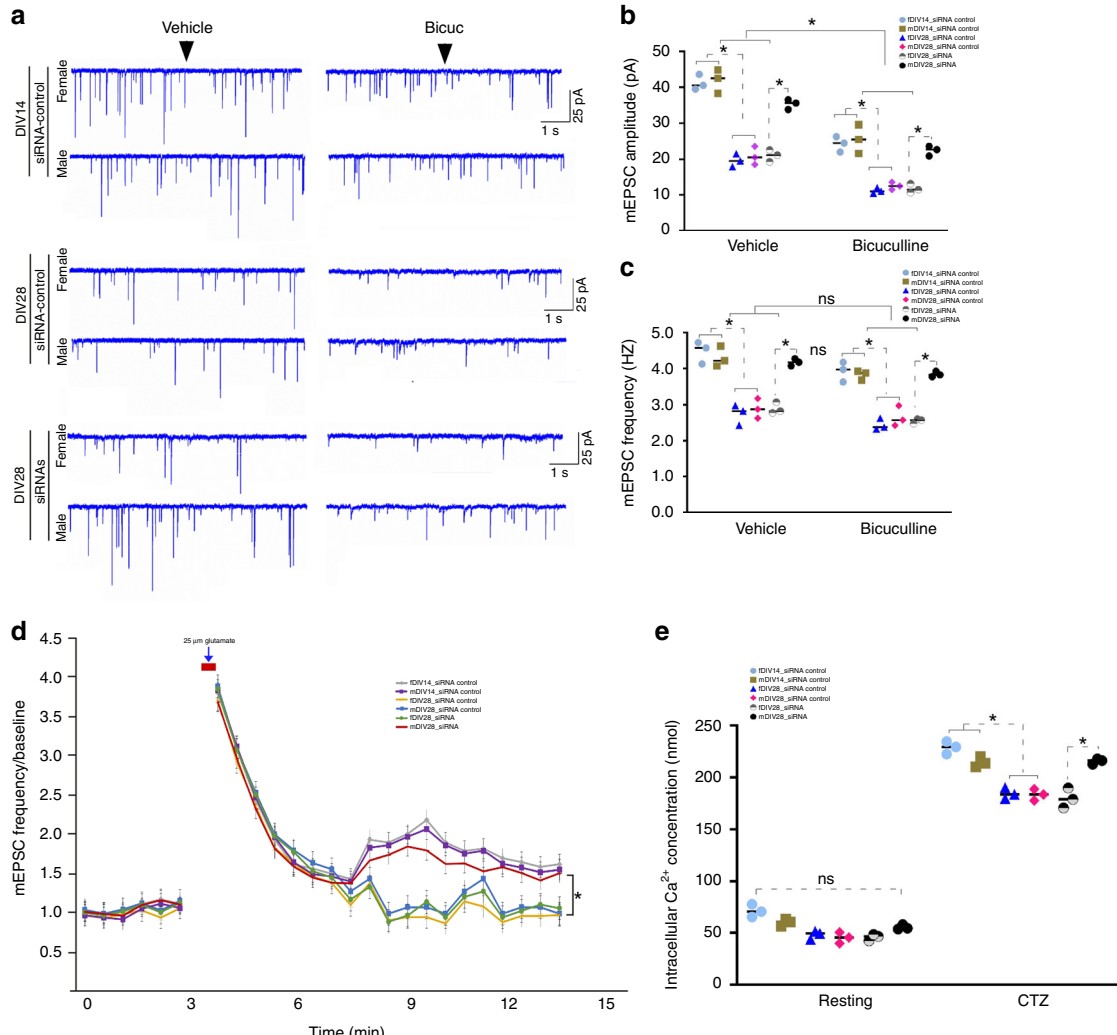

**Fig. 7 CircGRIA1 regulates synaptic plasticity and calcium homeostasis. a** Representative of mEPSC from hippocampal neurons. For bicuculline treatment, the neurons were incubated with bicuculline for 48 h followed by recording. Notes: fDIV14 and mDIV14, fDIV28 and mDIV28, hippocampal cultures of female and male fetal *macaques* at DIV14 and DIV28, respectively. **b, c** The average mEPSC wave forms and quantification of mEPSC amplitudes and frequency. The histogram showing a significant decrease in mEPSC amplitudes and frequency at DIV28 (*$p$ = 0.00236 for amplitude, and *$p$ = 0.00237 for frequency). Knockdown of circGRIA1 caused a significant increase in mEPSC amplitudes and frequency at DIV28 (*$p$ = 0.00362 for amplitude, and *$p$ = 0.00437 for frequency). Bicuculline treatment led to a significant decrease in mEPSC amplitudes. ANOVA $F_{7, 79}$ = 13.594 *$p$ = 0.0002, Tukey's multiple comparisons DIV14 control vs. bicuculline, *$p$ = 0.026; siRNA-control DIV28 control vs. bicuculline, *$p$ = 0.023; siRNA DIV28 control vs. bicuculline, *$p$ = 0.026 (n.s. no significance, $n$ = 6–8 recordings of each culture from 2 to 3 fetal *macaques* per sex group). **d** Involvement of circGRIA1 in essential properties of cLTP. Data are presented as mean ± S.E.M.; *$p$ < 0.05 vs. basal (ANOVA, Tukey post hoc test, $n$ = 8–10 neurons of each culture from 2 to 3 fetal *macaques* per sex group). **e** CircGRIA1 regulates calcium flux. Relative ratios of fluorescence at an emission frequency of 510 nm and excitation frequencies of 340 and 380 nm was collected and analyzed. Data are present as mean ± S.E.M. ($n$ = 8–10 cells of each culture from 2 to 3 fetal *macaques* per sex group). *$p$ < 0.05, unpaired $t$-test. $p$ values were calculated using two-tailed Student's $t$ test or one-way ANOVA with Sidak's correction and two-way ANOVA with Turkey's correction for multiple comparisons. Source data are provided as a Source Data file.

sequenced for confirmation. For the overexpression of circGRIA1, lentiviral plasmid with inserts coding three exons of circGRIA1 (with 5′-AG and 3′-GT included in the insertion) was used. The 5′UTR sequence (~330 bp) of *macaque* GRIA1 containing circGRIA1 binding site was amplified and cloned into pGL4.11 vector. To construct the overexpressing circGRIA1, the cDNA of circGRIA1 was cloned and inserted into the pLCDH-ciR, Tet-on circRNA, and AAV-hSyn1 expression vectors with cyclization sequence[45,46].

**Animals and samples collection.** Frozen postmortem tissue samples from *rhesus macaque* were obtained from Kunming Primate Research Center of the Chinese Academy of Sciences (KPRC). Brain regions were systematically collected from well-characterized rhesus monkeys born and raised at the KPRC in outdoor, 6-acre enclosures that provide a naturalistic setting and normal social environment. Extensive health, family lineage, and dominance information were maintained on all animals. For circRNA-seq analysis, two male and two female specimens at each

of stages representing adult (10-year old) and old (20-year old) were profiled (GEO, GSE94027)[26]. According to a widely used *macaque* brain atlas and brainmaps (http://www.Brainmaps.org), tissues spanning eight anatomically distinct regions were selected and collected from each specimen. The detailed information was described as below: the PFC was sampled at the main sulci, the posterior cingulate cortex was sampled at the Brodmann's area 23, the temporal cortex at the superior temporal gyrus, the parietal cortex at the middle sylvian fissure, the occipital cortex at the V1, and the cerebellar cortex was sampled at the cauda cerebellum. The hippocampus (including CA1 and DG) was also sampled. All the collected samples were washed with RNA later solution (AM7021, Ambion, USA) and put in freezing tubes to store at liquid nitrogen temperature.

All animal procedures were in strict accordance with the guidelines for the National Care and Use of Animals approved by the National Animal Research Authority (China) and the Institutional Animal Care and Use Committee of the Kunming Institute of Zoology of Chinese Academy of Sciences. The nonhuman primate cares and experimental protocols were approved by the Ethics Committee

of Kunming Institute of Zoology and the KPRC, Chinese Academy of Sciences (AAALAC accredited), and the methods were carried out in accordance with the approved guidelines (Approval No: SYDW-20140311).

**Conservation analysis for *macaque* circRNAs.** To analyze the homologous feature between *macaque* circRNAs and other mammalian species, we downloaded the human and mouse circRNAs sequence from the circBase (http://www.circbase.org/) {Glazar et al.[47] #56}. Then we aligned the circRNA sequence to the human and mouse circRNAs by blastn with $E$ value 1e − 3. Meanwhile, we also adopted the conservation calculation method in Rybak-Wolf et al.[23] {Rybak-Wolf et al.[23] #2}, and performed this analysis among three species: *macaque*, mouse, and human.

**Analysis of the relationships between circRNAs and host mRNAs.** To explore the expression relationship of circRNAs and their host mRNAs, we used the circRNA-seq data (GEO, GSE94027) {Xu et al.[26] #26} and the poly(A) selected RNA-seq data (GEO, GSE85377) {Liu et al.[48] #48}. The biological samples of the two studies were the same and total RNAs were extracted from the brain tissue parallelly. Thus we could compare the expression level between circRNAs and host mature mRNAs. Based on the expression of each circRNA and host mRNA, Pearson correlation coefficient (PCC) and $p$ value were obtained for each mRNA–circRNA pair. Then we filter the result by a given threshold, absolute PCC no less than 0.3 and $p$ value no more than 0.1. Besides the positive correlation pairs, negative pairs with correlation coefficient <0 were also included.

**Fetal *rhesus macaque* neural cell cultures.** Fetal *rhesus macaque* neuronal cultures were prepared and maintained according to the established protocol with modification {Negishi et al.[49] #131}. Briefly, fetal brains were obtained from second trimester fetal *macaques* (12–15 weeks) (KPRC of the Chinese Academy of Sciences) obtained via caesarian section with the verification of sex identity. Animals were housed in accordance with standards of the nonhuman primate cares and experimental protocols were approved by the Ethics Committee of Kunming Institute of Zoology and the KPRC, Chinese Academy of Sciences (AAALAC accredited), and the methods were carried out in accordance with the approved guidelines. The meninges were removed, and cortical tissue was mechanically dissociated using sterile scalpel blades and washed in sterile phosphate-buffered saline (PBS). Cells were pelleted by centrifugation at $1200 \times g$ for 10 min, and the pellet was subsequently enzymatically dissociated with 0.25% trypsin in the presence of DNase (50 μg/ml) at 37 °C for 30 min. Cells were further mechanically dissociated through a 150-μm pore size nylon mesh filter and washed again in cold sterile PBS by centrifugation at $2200 \times g$ at 4 °C for 10 min. The collected cell pellet was mechanically dissociated by trituration and further enzymatically dissociated with DNase (50 μg/ml) at RT for 10 min. After several further washes with cold sterile PBS, cells were seeded onto poly-D-lysine-coated glass coverslips or T75 flasks at a density of $1 \times 10^6$ cells/ml in DMEM/10% FCS. FDU (10 μM), a uridine analogue that is toxic to dividing astrocytes, was added to cultures to inhibit over proliferation of glial cells on days 5, 8, and 12 in vitro. Neurons were collected and prepared for biochemistry, immunocytochemistry, electrophysiology, and calcium imaging assays at indicated time.

**RNA extraction and reverse transcription PCR.** RNA was using PureLink micro-to-midi total RNA purification system (Invitrogen). RNA quantity and quality were evaluated by Nano Drop ND-1000 spectrophotometer (Nano Drop Thermo, Wilmington, DE) and RNA integrity was assessed by agarose gel electrophoresis. Specific divergent primers were designed to amplify the circular and linear *Rhesus macaque* transcripts. Semi-quantitative RT-PCR was performed with Superscript III one-step RT-PCR system with platinum Taq High Fidelity (Invitrogen). For quantitative real-time PCR, cDNAs were prepared by using oligonucleotide (dT), random primers, and a Thermo Reverse Transcription kit (Signal way Biotechnology). qPCR was performed with SYBR Green I Master (Roche, 04707516001) on Light Cycler 480 II. qPCR was performed in 10-μl reaction volume, including 2 μl of cDNA, 5 μl 2× Master Mix, 0.5 μl of Forward Primer (10 μM), 0.5 μl of reverse primer (10 μM), and 2 μl of double distilled water. The reaction was set at 95 °C for 10 min for pre-denaturation, then at 95 °C for 10 s, and at 60 °C for 60 s repeating 40 cycles. *Gapdh* was used as a reference. Both target and reference were amplified in triplicate wells. And the relative level of each circular and linear transcript was calculated using the $2^{-\triangle\triangle Ct}$ method. All PCR primer sequences can be found in Supplementary Data 3.

**Subcellular fractionation.** Cytoplasmic and nuclear fractional RNAs were extracted and prepared from postmortem fresh hippocampal tissues of 20-year-old male *macaques* according to the instruction of the Cytoplasmic and Nuclear RNA Purification Kit (Norgen). Extracted RNAs were subjected to RT-qPCR analysis to verify the subcellular localization of circGRIA1 with β-actin as the cytoplasm control and U6 as the nucleus control.

**BASEscope assay for circRNA detection.** BASEscope assays were performed using BaseScope™ Detection Reagent Kit-RED (#322900-USM, Advanced Cell Diagnostics (ACD) Hayward, CA) and circGRIA1 junction site-targeting

or -nontargeting labeled probes conjugated to HRP from ACD (#700001 and #700002). Briefly, 10-μm cryostat *macaque* brain sections were pretreated with hydrogen peroxide followed by performing target retrieval using an Oster® Steamer. Dried slides were placed on the slide rack, and incubated with RNAscope® Protease III at 40 °C for 30 min in the HybEZ™ system (ACD). Next, the slides were incubated at 40 °C in order to hybridize probes (circGRIA1) for 2 h in HybEZ™ system. The slides were then performed signal amplification with the following steps: AMP 0 for 30 min; AMP 1 for 15 min; AMP 2 for 30 min; AMP 3 for 30 min; AMP 4 for 15 min; AMP 5 for 30 min; AMP 6 for 15 min. After each step, slides were washed with wash buffer three times at room temperature (RT). Chromogenic detection was performed using BASEscope™ Fast RED followed by counterstaining with hematoxylin (American MasterTech Scientific, Lodi, CA). All images were collected using a Zeiss Olympus IX-81 microscope with either a 40× or 100× objective running Metamorph. For image analysis, regions of interest (50 × 50 μm) were manually drawn, and after background subtraction, BASEscope intensity was normalized to the control. Images were thresholded in Image J and, using the Image Calculator tool, a third image was generated showing those pixels which were positive in all input channels. Using the Particle Analysis tool, the size and number of the thresholded clusters were analyzed. Microsoft Excel was used to calculate the fraction of positive clusters. GraphPad Prism was used to perform ANOVAs and $t$-tests and to visualize bar charts. Error bars represent S.D. The target genes, probed regions, and sequences of target probes are listed in Supplementary Table S1.

**RNAscope assay for *Gria1* mRNA detection.** Detection of *macaque* *Gria1* was performed on cryostat brain sections of PFC and hippocampus of *rhesus macaque*s using RNAscope® Probe- Hs-GRIA1 (Cat No. 472441, ACD) and RNAscope® 2.5 LS Multiplex Fluorescent Reagent Kit v2 ASSAY (323100-USM, ACD). Positive [RNAscope® 3-plex LS Multiplex Control Positive Probe—Mm polr2A, PPIB, ubc; ACD] and negative [RNAscope® 3-plex LS Multiplex Negative Control Probe DAPB; ACD] controls were performed in parallel. Slides were thawed at RT for 10 min before baking at 60 °C for 45 min. The sections were then post-fixed in pre-chilled 4% PFA for 15 min at 4 °C, washed in 3 changes of PBS for 5 min each before dehydration through 50%, 70 and 100% and 100% ethanol for 5 min each. The slides were air-dried for 5 min before loading onto a Bond Rx instrument (Leica Biosystems). Slides were prepared using the frozen slide delay prior to pretreatments using Epitope Retrieval Solution 2 (Leica Biosystems) at 95 °C for 5 min, and ACD enzyme from the Multiplex Reagent kit at 40 °C for 10 min. Probe hybridization and signal amplification was performed according to manufacturer's instructions. The following TSA plus fluorophores were used to detect corresponding RNAscope probes using the BondRx platform according to the ACD protocol: Fluorescein (Akoya Biosciences), and Cy5 (Akoya Biosciences) were slides that were then removed from the Bond Rx and mounted using Prolong Diamond (ThermoFisher Scientific). Slides were imaged on a CellDiscoverer 7 microscope (Zeiss). *Gria1* positive cells were detected using the HALO FISH v2.1.6 analysis module based on intensity thresholds set using negative controls for both the fluorescein and Cy5 channels. Neurons detected as positive for *Gria1* were checked by eye, and were only included in final analysis if there were three or more spots corresponding to *Gria1* mRNA. The target genes, probed regions, and sequences of target probes are seen in the product information of RNAscope® Probe- Hs-GRIA1(Cat No. 472441, ACD).

**Dual DNA–RNA fluorescent in situ hybridization (FISH).** CircGRIA1 RNA probe was transcribed by the TranscriptAid T7 High Yield Transcription Kit (Thermo Scientific), with the corresponding insertion of circGRIA1 junction site in the T vector as a template for transcription, and were labeled with Alexa Fluor647 with the ULYSIS Nucleic Acid Labeling Kit (Invitrogen), which added a Fluor on every G of the probe to amplify the fluorescence intensity. Fixed neurons were washed in PBS, treated with RNase R at 37 °C for 30 min, and then neurons and circGRIA1 RNA probe were denatured at 80 °C for 15 min and then incubated at 42 °C for 24 h with human Cot-1 DNA (Life Technologies, final concentration 30 ng/μl). Slides were washed with 2× SSC at 45 °C for 10 min. FISH signal for circGRIA1 was detected with junction probe if not specified. For DNA–RNA double FISH, DNA probes were prepared with *Gria1* and *Gapdh* genomic PCRs and labeled with Alexa Fluor488 with a ULYSIS Nucleic Acid Labeling Kit; then hybridization was performed with the same conditions as for RNA FISH. Neurons were counterstained with 1-μg/ml DAPI. The primer sequences of target probes were listed in Supplementary Data 3.

**Immunohistochemistry (IHC).** For DAB/bright field staining, ten-micron cryostat sections of *macaque* brain were pretreated in 0.3% hydrogen peroxide in methanol for 30 min to remove endogenous peroxidase activity, rinsed in Tris-buffered saline (TBS), and then treated with 0.1-M citrate buffer in a microwave at sufficient power to keep the solution at 100 °C for 20 min. Sections were cooled in the same buffer at RT for 30 min and rinsed in TBS. Slides were incubated in 10% goat serum in PBS blocking solution for 1 h at RT, after which primary antibodies GluR1 or GluR2 were applied to the sections that were then incubated at 4 °C overnight. The sections were washed three times in TBS before applying the secondary antibody (Vector Laboratories). Secondary antibody was applied for 1 h at

RT. Afterwards, sections were rinsed three times in TBS. Rinsed sections were then incubated in Vectastain ABC Elite reagent for 1 h and developed using diamino-benzidine (Vector Laboratories). After dehydration all sections were mounted in Permount under a glass coverslip. Control sections were subjected to the identical staining procedure, except for the omission of the primary antibody.

**Protein extraction and western blot**. Male frozen *macaque* brain tissues or cultured fetal hippocampal neurons were lysed using 1× RIPA buffer (Sigma Aldrich), complemented with Protease Inhibitor Cocktail (PIC, Cat #11836153001, Roche) and Phosphatase Inhibitor Cocktail (Cat #11836153001, Roche). After protein quantification, samples were boiled for 10 min in SDS loading buffer (1:1 ratio). An aliquot (up to 50 μg) of the resulting sample was run on an SDS–PAGE gel and then transferred to a Hybond-PVDF membrane (Amersham Pharmacia). The membrane was blocked in 5% milk for 1 h at RT and incubated overnight with the appropriate primary antibody resuspended in 5% milk in TBST at 4 °C. Following three-time washes using TBST, the membrane was incubated for 2 h at RT with the appropriate secondary antibody resuspended in 5% milk in TBST at RT, followed by four washes using TBST. Signal ECL (Pierce) amplification was detected by Tanon 5200 multi chemiluminescent image system (Tanon, Shanghai, China). Signal intensity was quantified with Image J (National Institute of Health).

**Dual-luciferase reporter assay**. Luciferase assays were performed using the Dual-Luciferase Reporter Assay System (Promega, Beijing, China). Briefly, SH-SY5Y cells grown in 48-well plates were transfected with the 5′UTR sequence (~330 bp) of *macaque Gria1* gene in pGL4.11 vector, *Renilla*, and circGRIA1 in Tet-on circRNA or pLCDH-ciR vector. Firefly and *Renilla* luciferase activities were measured 24 h after transfection. The light intensity from the firefly luciferase was normalized to *Renilla* luciferase.

**Immunofluorescence**. The *rhesus macaque* brain cryostat sections were first rinsed in PBS, followed by pretreatment in antigen unmasking solution (low pH) for 30 min at 100 °C. After the slides had cooled in buffer for 30 min at RT, slides were rinsed in PBS. Sections were incubated in 10% goat serum in PBS to block non-specific binding for 1 h at RT, followed by incubation with the primary antibody against PSD95 (Abcam, ab13552) and Synapsin-I (Abcam, ab64581) overnight at 4 °C. Primary antibodies were visualized with the appropriate Alexa-Fluor488 and Alex-Fluor 594 antibodies (Invitrogen). All sections were mounted in ProLong Gold antifade reagent with DAPI (Invitrogen) under a glass coverslip. All experiment was conducted at least in triplicates. For synaptic puncta analyses, cells were observed with a confocal laser scanning microscope (Nikon).

For primary cultures, fetal *macaque* hippocampal neurons were rinsed once with PBS and then fixed in buffered 4% paraformaldehyde in 0.1-M phosphate buffer for 30 min at RT followed by three rinses with PBS. Primary antibody GluR1 concentrations used for cell culture were 1:200. All secondary antibodies were used at 1:1000. Cells were counterstained with 1-μg/ml DAPI. All immunofluorescence images were collected using a Zeiss Olympus IX-81 microscope with either a 40× or 100× objective running Metamorph. Microsoft Excel was used to calculate the fraction of positive clusters. GraphPad Prism was used to perform *t*-tests and to visualize bar charts. Error bars represent S.D.

**DIG-labeled antisense RNA probes synthesis by in vitro transcription**. PCR primers were designed using standard primer designing tools (Primer Premier 5.0) to amplify 100–300 nt fragment corresponding to linear and circular RNA sequences or 100–150 nt fragment corresponding to head-to-tail junction (short circRNA specific probes and not overlapping with linear RNA sequence). T7 promoter sequence was added to the reverse primer to obtain an antisense probe in vitro transcription reaction. In vitro transcription was performed using the TranscriptAid T7 High Yield Transcription Kit (K0441, Thermo Scientific) with DIG-RNA labeling mix (Roche) according to manufacturer's instruction. DNA templates were removed by DNAs I digestion and RNA probes purified by phenol chloroform extraction and subsequent precipitation. Probes were used at 50 ng/ml final concentration (Northern blot). The primer sequences of probes were seen in Supplementary Data 3.

**Northern blots for circRNA detection**. Total RNA (10 μg for 10- and 20-year old male *macaque* brain samples, 2 μg for fetal *macaque* hippocampal primary neurons) was denatured using NorthernMax®-Gly sample loading dye (Ambion) and resolved on 1.2% agarose gel in MOPS buffer. The gel is soaked in 1×TBE for 20 min and transferred to a Hybond-N+ membrane (GE Healthcare) for 1 h (15 V) using a semi-dry blotting system (Bio-Rad). Membranes were dried and UV-crosslinked with 150 mJ/cm² at 254 nm. Pre-hybridization was done at 68 °C for 1 h, and using the DIG Northern Blot Starter Kit (12039672910, Roche). DIG-labeled in vitro transcribed *Gria1* junction site-targeting circular, and junction site-nontargeting linear probes were hybridized overnight. The membranes were washed three times in 2 × SSC, 0.1% SDS at 68 °C for 30 min, followed by three 30 min washes in 0.2 × SSC, 0.1% SDS at 68 °C. The immunodetection was performed with anti-DIG AP-conjugated antibodies. Immunoreactive bands were visualized using CDP star reagent (Roche) and a LAS-4000 detection system (GE Healthcare). The primer sequences of probes were seen in Supplementary Data 3.

**siRNA knockdown in fetal *macaque* brain cultured neurons**. The high lentiviral particle titers were prepared with 3rd Generation Packaging Mix in HEK 293T cells and purified by Ultra-Pure Lentivirus purification Kits (LV998, Applied Biological Material Inc. Vancouver). For the knockdown assay, after 5 days in culture, fetal *macaque* hippocampal neurons were infected using a multiplicity of infection (moi) between 5 and 10 to provide an efficiency of infection above 70%, which piLen-ti™-GFP was used a co-expressed reporter to monitor infection. Samples were collected at indicated time and different assays were performed. All sequences of siRNA against circRNAs can be found in Supplementary Table 1.

**Chromatin isolation by RNA purification (CHIRP)**. CHIRP or biotin-oligonucleotides pulldown was carried out according to the established protocol with modifications[50]. Briefly, fresh *macaque* brain tissues digested by 0.25% trypsin or fetal *macaque* hippocampal neurons were crosslinked with 1% formaldehyde in PBS for 10 min at RT, and cross-linking was then quenched with 0.125-M glycine for 5 min. The tissues or neurons were pelleted and resuspended in swelling buffer (0.1 M Tris, pH 7.0, 10 mM KOAc, and 15 mM MgOAc, with freshly added 1% NP-40, 1 mM DTT, complete protease inhibitor, and 0.1 U/μl RNase inhibitor) for 10 min on ice. Cell suspensions were then homogenized and pelleted at 2500 g for 5 min. Nuclei were further lysed in nuclear lysis buffer (50 mM Tris, pH 7.0, 10 mM EDTA, and 1% SDS; with freshly added 1 mM DTT, complete protease inhibitor, and 0.1 U/μl RNase inhibitor) on ice for 10 min and were sonicated until most chromatin had solubilized and DNA was in the size range of 100–500 bp. Chromatin was diluted in two times volume with hybridization buffer (750 mM NaCl, 1% SDS, 50 mM Tris, pH 7.0, 1 mM EDTA, 15% formamide, 1 mM DTT, complete protease inhibitor, and 0.1 U/μl RNase inhibitor). Biotin-DNA oligos (100 pmol) were prepared with the ULYSIS Nucleic Acid Labeling Kit (MP 21650, Invitrogen) and added to 3 ml of diluted chromatin, which was mixed by end-to-end rotation at 37 °C for 4 h. M-280 Streptavidin Dynabeads (Life Technologies) were washed three times in nuclear lysis buffer, which was blocked with 500 ng/μl RNA and 1 mg/ml BSA for 1 h at RT, then washed three times again in nuclear lysis buffer before being resuspended. 100 μl Dynabeads were added per 100 pmol of biotin-antisense oligos, and the whole mix was then rotated for 30 min at 37 °C. Beads were captured by magnets (Life Technologies) and washed five times with 40× the volume of Dynabeads with wash buffer (2× SSC, 0.5% SDS, and 0.1 mM DTT and PMSF (fresh)). Beads were then subjected to RNA elution and DNA elution. The sequences of probes for CHIRP were seen in Supplementary Table 2.

**Preparation of nascent RNAs**. Nascent transcripts were isolated by fluorescent labeling of nascent RNA by use of the Click-iT Nascent RNA Capture Kit (ThermoFisher Scientific Inc.). Briefly, fetal *macaque* hippocampal neurons at indicated DIV were incubated for 24 h in medium (see above) containing 0.5-mM 5-ethynyl uridine (EU) at 37 °C, 5% CO₂, 100% humidity. After incubation, neurons were once washed in PBS followed by proceeding to RNA extraction using TRIzol. Next, biotinylation of RNA by the Click-iT reaction was performed. The biotinylation of EU-labeled RNA was precipitated and proceeded to bind biotinylated RNA to Dynabeads® MyOne™ Streptavidin T1 magnetic beads. The cDNA synthesis using the RNA captured on the beads as a template was immediately performed. The cDNA was stored at –20 °C until further use.

**MRI-guided AAV viral particles microinjection in vivo**. Three-paired 10- and 20-year-old male and female *rhesus macaque*s weighing 6–10 kg were used in microinjection of the viral particles. The monkeys were housed in adjoining individual primate cages under typical conditions of humidity, temperature, and light and fed with standard monkey chow and daily supplements of fruit and vegetables to ensure their health and welfare. Animals were fasted overnight prior to the MRI sessions and surgery. 3.0 T MRI (uMR770, United Imaging) scanning was conducted to determine the microinjected location of hippocampus with a circular 12-chanel coil. Before the scan, each *macaque* was sedated with ketamine (10 mg/kg, i.m.) and atropine (0.05 mg/kg, i.m.), anesthetized with pentobarbital (25 mg/kg, i.m.) and then placed into the scanner in the prone position. The whole-brain images were acquired with a 3D T1-weighted sequence (TR = 10.6 ms, TE = 3.6 ms, TI = 0 ms, NEX = 2, slice thickness = 0.5 mm, matrix size = 256 × 256, FOV = 124 × 124 mm) and a 3D T2-weighted sequence (TR = 2300 ms, TE = 380.88 ms, TI = 0 ms, NEX = 2, slice thickness = 0.5 mm, matrix size = 256 × 256, FOV = 124 × 124 mm). In order to design and optimize the target trajectory, all the MRI images were analyzed by a commercial neuronavigation system (Brainsight, Rogue Research). One hemisphere received five intrahippocampal injections (Bregma −9.5/−12.6/−12.6/−14/−18 mm, ML +12.45/+9.5/+13.5/+14.8/+15 mm, DV 34/34/34/33.5/30 mm) of 20 μL (total 100 μL) regulative AAV vectors (serotype AAV9 and titer ~3 × 10¹⁴ v.g./mL), and the contralateral hemisphere (Bregma −9.5/−12.6/−12.6/−14/−18 mm, ML −12.5/−9.5/−13.5/−14.8/−15 mm, DV 34/34/34/33.5/30 mm) received control vectors (serotype AAV9 and titer ~3 × 10¹⁴ v.g./mL). All surgical procedures were conducted under strict aseptic conditions. After the skull was exposed, holes were punched in the corresponding positions and the vectors were then infused through a 31-gauge Hamilton syringe placed in a syringe pump (Stoelting Apparatus) attached to a stereotaxic instrument. A map of the microinjection sites is shown in Supplementary Fig. 7 according to the Macaque Scalable Brain Atlas with modifications (Author copy:

arXiv:1312.6310) {Bakker, 2015 #175; Calabrese, 2015 #174}. The viral particles (20 μL) were infused at a rate of 500 nL/min. Following infusion, the needle was left in place for 10 min before being slowly retracted from the brain. As a prophylactic antibiotic treatment, cephalosporin was injected for three consecutive days after surgery (25 mg/kg/day, i.m., once a day). Six weeks after injection, the monkeys were anesthetized with pentobarbital and transcardially perfused with 1 L of cold PBS. The brains were then removed rapidly from the skull and divided by the *macaque* brain mold. Fresh tissues were sampled and stored in freezing tubes at liquid nitrogen temperature. The remaining tissues were prepared and fixed with 4% paraformaldehyde. The cryostat sections of hippocampus were prepared for IHC, BASEscope, and RNAscope ISH.

**Electrophysiology**. Whole-cell patch-clamp recordings were made with an Axopatch 200B amplifier from hippocampal cultures of fetal *macaque* (DIV14 and DIV28) bathed in HBS containing 119 mM NaCl, 5 mM KCl, 2 mM CaCl$_2$, 2 mM MgCl$_2$, 30 mM glucose, 10 mM HEPES [pH 7.4; ~310 mOsm] plus 20 μM CNQX. Whole-cell pipette internal solution contained 120 mM potassium gluconate, 20 mM KCl, 0.1 mM EGTA, 2 mM MgCl$_2$, 2 mM adenosine triphosphate, 0.4 mM guanosine triphosphate, 10 mM HEPES (pH 7.2; ~300 mOsm) and the pipette resistances ranged from 4 to 6 MΩ. 20 μM bicuculline was added in conditioned media for 48 h and the media was replaced with the HBS 15 min prior to recording. Neurons were voltage clamped at −70 mV while the series resistance was left uncompensated during the recordings. mEPSCs were analyzed offline using Stimfit software by employing a template-matching algorithm. Recordings were started 5 min after patching and the recording duration usually ranged from 5 to 10 min. Only recording epochs in which series and input resistances varied <10% were analyzed. At least ten neurons were subjected to recording for each group. Statistical comparisons were made using unpaired student's *t* test; for multiple comparisons single factor ANOVAs followed by nonparametric Mann–Whitney *U*-test were performed.

**Chemically induce LTP (cLTP)**. Whole-cell voltage-clamp recordings were obtained from hippocampal cultures of fetal *macaques* at DIV14 and DIV28. Currents were recorded from neurons held at −70 mV with a Multiclamp 200B amplifier (Molecular Devices, Sunnyvale, CA) and digitized by Digidata 1440A (Molecular Devices, Sunnyvale, CA). pCLAMP software (Molecular Devices, Sunnyvale, CA) was used for data acquisition and analysis. Neurons were transferred to RT and were superfused with standard extracellular solution, 300 mOsm, pH 7.4, containing 140 mM NaCl, 2.5 mM KCl, 10 mM HEPES, 20 mM glucose, 2 mM CaCl$_2$, 4 mM MgCl$_2$, and 1 μM tetrodotoxin. The patch pipette (4-6 MΩ resistance) internal solution, 290 mOsm, pH 7.2, contained 115 mM potassium gluconate, 20 mM KCl, 10 mM HEPES, 1 mM EGTA, 2 mM MgCl$_2$, 4 mM ATP-Na$_2$, 0.4 mM GTP-Na, and 10 mM phosphocreatine. Neurons were exposed to Mg$^{2+}$ free extracellular solution for a few seconds. Then, glutamate-induced LTP was induced by brief (30 s) challenge with exogenous L-glutamate (25 μM; in Mg$^{2+}$ free standard extracellular solution), followed by standard extracellular solution.

**Calcium flux analysis**. For calcium flux analysis, hippocampal cultures from fetal *macaque* were prepared as described above, seeded onto 110-μm-thick coverslips, and cultured as above. Cells were washed with PBS and loaded with 5-μM Fura-2 (Molecular Probes, Eugene, OR) for 1 h in a dark chamber at 37 °C. Cells were then washed with PBS, and DMEM/1% FCS was added to the cultures. Neurons were kept at 37 °C until analyzed for calcium flux responses (up to 1 h). For calcium flux analysis, coverslips were placed into a 37 °C warming chamber and examined under an inverted microscope connected to a spectrofluorometer. Fetal *macaque* neurons with infection of the junction site-targeting siRNAs against circGRIA1 or the mismatched junction site-targeting siRNA-control, respectively, were tested for response to CTZ. Neurons were stimulated with 25-μM glutamate before exposure to 100-μM CTZ. Calcium flux tracings were analyzed for the maximum increase in intracellular calcium according to the formula $[Ca]_i = k_d [(R − R_{min}/R_{max} − R)]$, assuming a $K_d$ of 224 nM, and $R$ is the ratio of fluorescence at 340 and 380 nM. Calcium concentrations are expressed as the mean ± S.D., and all data were presented as the relative ratio of fluorescence at an emission frequency of 510 nm and excitation frequencies of 380 nm and are representative of three separate experiment per treatment with five different neuronal preparations.

**Statistical analysis**. Two-sided paired *t*-test was performed to calculate the differently expressed circRNAs. The statistical significance of the data of BASEscope and RNAscope ISH, and IHC was tested using either an unpaired *t*-test or multiple comparisons single factor ANOVAs since the normality of the distribution was pretested using the Lilliefors test. No statistical methods were used to predetermine sample sizes, but our sample sizes are similar to those generally employed in the field. Data collection and analysis were not performed blind to the conditions of the experiment and no randomization of data was performed.

**Reporting summary**. Further information on research design is available in the Nature Research Reporting Summary linked to this article.

## Data availability

All data and genetic material used for this paper are available from the authors on request. The circRNA-seq data have been deposited in GEO and are accessible through accession number GSE94027. The poly(A)-enriched RNA-seq data has been deposited in GEO and are accessible through accession numbers GSE85377 (https://www.ncbi.nlm.nih.gov/geo/query/acc.cgi). Source data are provided with this paper.

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

## Acknowledgements

We are grateful to Ling-Ling Chen from Shanghai Institute of Biochemistry and Cell Biology for granting Tet-on circRNA and AAV-hSyn1 expression vector, and Geenseed Inc. (Guangzhou, China) for pLCDH-ciR vector. We also thank Dong Chen and Yi Zhang from Ablife Inc. for their kind help in the performance of the bioinformatics. We also thank Huanzhi Chen and Nanhui Chen for their help on MRI scanning and operation of *macaque* brain. The authors also thank Karl Herrup and Hei-Man Chow (the Hongkong University of Science and Technology) for thoughtful comments and editorial assistance. This work was supported by the National Natural Science Foundation of China (NSFC 91649119) to J.L., the Ministry of Science and Technology of China (2015CB755605) to J.L., and Peking University (BMU2019YJ001) to J.L.

## Author contributions

K.X. and J.L. conceived and designed all experiments and interpreted results. K.X. and J. L. performed bioinformatic analysis; K.X., Y.Z., and W.X. performed Northern blot, IHC, IF and BASEscope ISH and RNAscope ISH, luciferase reporter assay, dual FISH, RNA pulldown, CHIRT, and RT-qPCR; Y.Z., K.X., Z.W., C.L., Z.H., and L.Lv. in vivo microinjection of viral particles into the brains of *macaques*, and fetal *macaque* hippocampal cultures; K.X. and Z.Z. performed electrophysiology, cLTP and calcium imaging, and Ca$^{2+}$ influx; K.X. and J.L. wrote the paper with assistance from Y.Z., W.X., Z.Z., Z. W., C.L., Z.H., L.Lv., Y.Z., L.L., and X.H. and J.L. supervised the project.

## Competing interests

The authors declare no competing interests.
