## [Peer Review File · Nature Communications]

Reviewers' comments:

Reviewer #1 (Remarks to the Author):

The authors report that circular RNA for the GluA1 AMPA subunit is reduced in specific brain regions of aged male, not female, rhesus monkeys. The reduction in circGRIA1 is well documented and it could become potentially a very interesting finding. Particularly because it was observed in primates. The exploration of the functional consequences of these changes in cultured fetal cells is not convincing, however. First the observation that manipulating circGRIA1 affects AMPA-mediated function is almost a foregone conclusion. Nice confirmation, but not surprising or revealing. Second, and more important, the changes occurring between the first and the second week in the dish might model embryonic development, but at no stretch of imagination could model aging changes occurring a decade later and taking a decade to develop.

Reviewer #2 (Remarks to the Author):

In this study, Xu et al. found that GluR1 gene-derived circGRIA1 regulates homeostatic synaptic plasticity in the biology of brain aging. Firstly, according to the RNA-seq data in their previous work, the authors focused on circGRIA1, which is upregulated in male rhesus macaque brain over the aging states. They identified a negative correlation between circGRIA1 and its host mRNA expression. Then, they found that circGRIA1 can interact with the 5'-UTR genomic region of its parental gene and negatively regulate host mRNA expression. In addition, circGRIA1 regulates synaptogenesis and AMPA receptor activity-related calcium homeostasis and is associated with homeostatic synaptic plasticity in male fetal rhesus macaque hippocampal neurons. There are some suggestions that authors can address to improve and strengthen their results below.

Major comments:

1)The authors found that circGRIA1 binds to the 5'-UTR genomic region of its parental gene and negatively regulates the transcriptional activity of its parental mRNA gene. However, the evidences here are not enough. There is another probability that circGRIA1 may promote GluR1 mRNA degradation. Further studies like actinomycin D treatment are necessary to prove whether circGRIA1 inhibits nascent mRNA or promotes mRNA degradation. Also, to reveal the functions of circGRIA1, the subcellular localization experiment needs to be done.

2)The authors claimed that circGRIA1 was involved in synaptogenesis and regulation of homeostatic synaptic plasticity via negatively regulating GluR1 expression. However, there is no experimental evidence in this study to prove the relationship between GluR1 and homeostatic synaptic plasticity. Some mutations, which block the binding ability of circGRIA1 to its host gene, can be designed to verify whether the homeostatic synaptic plasticity regulation function is due to negatively regulating GluR1.

3)The authors detected multiple isoforms of circRNAs derived from GluR1. A graphical presentation containing the back-splicing junction sites of circRNAs, the numbers of reads that cover the back-splicing junction sites is necessary here to directly show us all these isoforms. How is the expression level of circGRIA1 compared to other GluR1 derived circRNAs?

4)The authors used the circGRIA1 isoforms in human and mouse from circBase and then calculated their conservation across the three species. However, a more recent study published the circRNA expression profiles from multiple tissues of human, monkey and mouse (Cell Reports, 2019, 26:3444-3460). The authors should use this new dataset to confirm their findings (conservation, expression level, etc) and particularly the expression landscape of this circGRIA1 in different tissues.

5) In the first section of Results, the authors used find_circ and CIRI2 to detect circRNA isoforms from the genomic locus of Gria1. Both tools should be cited here. In addition, the coordinates of these isoforms and their experimental confirmation should be provided as well.

6) There are some mistakes in figures and figure legends.

Fig 1. In the legend of (d), "the specific Digoxin-labeled" (Line 954) is repeated.

Fig 2. Where is the legend of (g)?

Fig 5. Where is the legend of (e)?

Fig 6. There are no figures correspond to fig.6d in line 275.

Minor comments:

1) Line 35. "with the 5'-UTR genomic loci of its parental gene" or "its parental gene's 5'-UTR genomic loci"

2) Line 39. and prevents... (no comma)

3) Line 70. when compared with...

4) Line 71. channels (plural)

5) Line 87. regulate (not "regulate to")

6) Line 94-95. Post-mortem brain tissues of 10- and 20-year old male macaque

7) Line 109. positive-correlated circRNAs

8) Line 122. frozen brain samples of 10- and 20-year old male macaque

9) Line 354-355. circRNAs are highly expressed in brain and testis compared with other tissues in mammals. Also, the reference 39 cited here can not support this opinion.

Reviewer #3 (Remarks to the Author):

Kaiyu Xu et al. describe a sex-specific regulation of gene production through the binding of a circRNA to its own genomic locus. The authors used brains and cultured fetal neurons to perform the experiments. The self regulation of the locus through the circRNA is dependent on age. As the major finding, the authors could link the differential expression of circGRIA1 to the alteration of neuronal plasticity and to the Alzheimer's disease. Although, the described mechanism is indeed highly interesting, the authors fail to back up their hypothesis with the presented data. Missing controls, incompletely described experiments, and statistical irregularities make this manuscript not suitable for publication.

Major concerns:

1. Sex specific controls

Kaiyu Xu et al. describe a male-specific regulation of gene expression and neuronal plasticity during aging. The authors show several controls for the presented experiments. However, the best and most valid control would be female samples, which should not display the described phenomena. It is essential for the manuscript to add the female control for figures 2 - 7, as well as for s2, s3, s5 - s8.

2. Nuclear mechanism of action

The authors base their hypothesis on the nuclear localization of circGRIA1. However, there is no solid evidence for this idea. Controversially, BASEscope staining presented in figures 2a, 2e, 4a, S7a show the localization of the majority of the circRNA outside of the nucleus and the fish experiment presented in S6a needs a Z-stack representation to be fully trusted. The reviewer finds it important to provide nuclear fractionation and demonstrate the purity of the samples via nucleus specific and cytoplasm specific genes.

3. Statistical evaluation

The description of the statistical analysis given by the authors is very often not clear. Moreover, the reviewer is especially concerned about quantification data of cells. In f1g c, e, S1b the authors speak of "n=3 repetition of the experiment". Does that mean that the biological n is 1? Further, the authors are referring to "independent experiments". How many individual animals were used per group and per "independent experiment"? It is of very high importance that the authors provide a clear explanation of the biological and technical replicates they used.

In figures 2, 4, s3, s6, s7, the authors counted the intensity of a marker within a cell. From the images presented in the figures, it is hardly possible to find the outline on one given cell. Since the authors used the program Image J for the quantification, the reviewer would ask to see the original images on which the quantification has been performed.

Minor concerns:

4. In f3a, the authors say "either 20-years old male macaque CA1 or fetal macaque hippocampal neurons..". So, what is it?
5. In f3d and e the authors compare two populations (10Y and 20Y) to one input. What is the input and how is this comparison valid?
6. Figures 1f and 3f are sequencing data of several experiments. Why did the authors not present a sequence alignment of the experiments instead of the pure sequence?
7. In figures 4a, b, d, and e, the authors are using scrambled siRNA and siRNA-control. What is the difference?
8. In f4a when OE, siRNA-control, and circGRIA1 siRNA demonstrate a much lower amount of nuclei. How do the authors explain these observations? The amount of red staining is reduced in siRNA-control, how can this be explained? Also, the quality of all the Gria1 fluorescence images is very bad. The reviewer strongly suggests to use more reliable images with a neuronal counterstaining and clearly visible DAPI.
9. In f4b and c the authors performed a t-test on three groups. How is this possible?
10. In f4d, the loading control for DIV28 siRNA-control and siRNA is strongly overlapping. How did the authors do quantification on such a blot? Also, since this blot has been repeated several times (as indicated in f4e), it is desirable to see all the blots that have been used for the quantification.
11. f5c,d and S8 c, d, demonstrate an increase of synapsin I. To validate the data, the authors have to present a western blot and its quantification.
12. The viral overexpression of circGRIA1 has to be validated by northern blot.
13. s3f shows the quantification of the western blot from s3e. Please provide all and the full western blots used for the quantification.
14. In s5a and b the authors compare two populations (DIV14 and DIV28) to one input. What is

the input and how is this comparison valid?

15. In s6a and b, the authors used FISH to detect the parental locus of circGRIA1. Since the locus is present in every cell, the authors have to explain why the signal is not clearly visible in every cell.

16. In s6a, the authors describe an experiment with Gapdh. However, Gapdh is not mentioned in the figure at all.

We appreciate the possibility to submit a revised version of our manuscript to *Nature Communications*. We are also grateful for the positive response and constructive feedback that we received from the referees. Their insightful comments helped us to substantially strengthen the manuscript and we hope that you will find it suitable for publication in *Nature Communications*. Please see below for our point-by-point response to the issues raised.

Reviewer #1 (Remarks to the Author):

The authors report that circular RNA for the GluA1 AMPA subunit is reduced in specific brain regions of aged male, not female, rhesus monkeys. The reduction in circGRIA1 is well documented and it could become potentially a very interesting finding. Particularly because it was observed in primates. The exploration of the functional consequences of these changes in cultured fetal cells is not convincing, however. First the observation that manipulating circGRIA1 affects AMPA-mediated function is almost a foregone conclusion. Nice confirmation, but not surprising or revealing. Second, and more important, the changes occurring between the first and the second week in the dish might model embryonic development, but at no stretch of imagination could model aging changes occurring a decade later and taking a decade to develop.

Response: We highly appreciate the referee's positive comments and constructive suggestions, which are truly helpful for the improvement of this manuscript. To address the limitation of *in vitro* cultural system raised by the referee, *in vivo* validation of circGRIA1 function has been investigated via microinjecting viral particles of siRNA against circGRIA1 into the hippocampus of macaque. The data provided solid evidence to support regulatory role of circGRIA1 in the biology of brain aging.

Reviewer #2 (Remarks to the Author):

In this study, Xu et al. found that GluR1 gene-derived circGRIA1 regulates homeostatic synaptic plasticity in the biology of brain aging. Firstly, according to the RNA-seq data in their previous work, the authors focused on circGRIA1, which is upregulated in male rhesus macaque brain over the aging states. They identified a negative correlation between circGRIA1 and its host mRNA expression. Then, they found that circGRIA1 can interact with the 5'-UTR genomic region of its parental gene and negatively regulate host mRNA expression. In addition, circGRIA1 regulates synaptogenesis and AMPA receptor activity-related calcium homeostasis and is associated with homeostatic synaptic plasticity in male fetal rhesus macaque hippocampal neurons. There are some suggestions that authors can address to improve and strengthen their results below.

Response: Thank the referee very much for your summary.

Major comments:

1)The authors found that circGRIA1 binds to the 5'-UTR genomic region of its parental

gene and negatively regulates the transcriptional activity of its parental mRNA gene. However, the evidences here are not enough. There is another probability that circGRIA1 may promote GluR1 mRNA degradation. Further studies like actinomycin D treatment are necessary to prove whether circGRIA1 inhibits nascent mRNA or promotes mRNA degradation. Also, to reveal the functions of circGRIA1, the subcellular localization experiment needs to be done.

Response: Thanks for your valuable suggestions and the suggested experiment has been done. We have performed the subcellular localization experiment (Supplementary Fig 1c) and luciferase reporter assay (Supplementary Fig 5c, d) to support the regulation of *Gria1* mRNA expression by circGRIA1 via *cis*-acting manner.

2)The authors claimed that circGRIA1 was involved in synaptogenesis and regulation of homeostatic synaptic plasticity via negatively regulating GluR1 expression. However, there is no experimental evidence in this study to prove the relationship between GluR1 and homeostatic synaptic plasticity. Some mutations, which block the binding ability of circGRIA1 to its host gene, can be designed to verify whether the homeostatic synaptic plasticity regulation function is due to negatively regulating GluR1.

Response: Thank you for your suggestion. While it is difficult to find a mutation to block the binding ability of circGRIA1 to its host gene at present stage, the luciferase reporter assay has been performed to reveal the association of circGRIA1 with the promoter region of *Gria1* genomic loci (Supplementary Fig 5c, d) plus previous data of CHIRP assay, suggesting negative regulation of *Gria1* mRNA expression by circGRIA1 via *cis*-acting way. Notably, *in vitro* and *in vivo* studies show knockdown of circGRIA1 not only increases GluR1 expression but also synapsin I level, both of which are involved in regulation of homeostatic synaptic plasticity. Physiological examination in hippocampal cultures further support this hypothesis.

3)The authors detected multiple isoforms of circRNAs derived from GluR1. A graphical presentation containing the back-splicing junction sites of circRNAs, the numbers of reads that cover the back-splicing junction sites is necessary here to directly show us all these isoforms. How is the expression level of circGRIA1 compared to other GluR1 derived circRNAs?

Response: This requested analysis has been performed (new Fig. 1a and Supplementary Table 1).

4)The authors used the circGRIA1 isoforms in human and mouse from circBase and then calculated their conservation across the three species. However, a more recent study published the circRNA expression profiles from multiple tissues of human, monkey and mouse (Cell Reports, 2019, 26:3444-3460). The authors should use this new dataset to confirm their findings (conservation, expression level, etc) and particularly the expression landscape of this circGRIA1 in different tissues.

Response: Thanks for your comments and suggestions. We have performed this analysis and showed the results. From our results, we found homologues from human and mouse

for circGRIA1, indicating its conservation in mammal. The homologous circRNAs were detected from brain and spinal cord tissues in circAtlas, with higher expression in spinal cord (Supplementary Table 1 and 2).

5) In the first section of Results, the authors used find_circ and CIRI2 to detect circRNA isoforms from the genomic locus of Gria1. Both tools should be cited here. In addition, the coordinates of these isoforms and their experimental confirmation should be provided as well.

Response: Thanks for your comments. We have cited these two tools. We have provided the coordinates of these circRNAs. Further, the relative levels of expression of all circGRIA isoforms have been showed in Supplementary Table 1.

6) There are some mistakes in figures and figure legends.

Fig 1. In the legend of (d), “the specific Digoxin-labeled” (Line 954) is repeated.

Fig 2. Where is the legend of (g)?

Fig 5. Where is the legend of (e)?

Fig 6. There are no figures correspond to fig.6d in line 275.

Response: Thanks for the referee’s careful reviewing. All of errors have been corrected.

Minor comments:

1) Line 35. “with the 5’-UTR genomic loci of its parental gene” or “its parental gene’s 5’-UTR genomic loci”

Response: Thanks for the referee’s careful reviewing. The error has been corrected.

2) Line 39. and prevents... (no comma)

Response: Thanks for the referee’s careful reviewing. The error has been corrected.

3) Line 70. when compared with...

Response: Thanks for the referee’s careful reviewing. The error has been corrected.

4) Line 71. channels (plural)

Response: Thanks for the referee’s careful reviewing. The error has been corrected.

5) Line 87. regulate (not “regulate to”)

Response: Thanks for the referee’s careful reviewing. The error has been corrected.

6) Line 94-95. Post-mortem brain tissues of 10- and 20-year old male macaque

Response: Thanks for the referee’s careful reviewing. The error has been corrected.

7) Line 109. positive-correlated circRNAs

Response: Thanks for the referee’s careful reviewing. The error has been corrected.

8) Line 122. frozen brain samples of 10- and 20-year old male macaque

Response: Thanks for the referee’s careful reviewing. The error has been corrected.

9)Line 354-355. circRNAs are highly expressed in brain and testis compared with other tissues in mammals. Also, the reference 39 cited here can not support this opinion.

Response: Thanks for the referee's careful reviewing. The error has been corrected, and the reference 39 has been removed.

Reviewer #3 (Remarks to the Author):

Kaiyu Xu et al. describe a sex-specific regulation of gene production through the binding of a circRNA to its own genomic locus. The authors used brains and cultured fetal neurons to perform the experiments. The self regulation of the locus through the circRNA is dependent on age. As the major finding, the authors could link the differential expression of circGRIA1 to the alteration of neuronal plasticity and to the Alzheimer's disease. Although, the described mechanism is indeed highly interesting, the authors fail to back up their hypothesis with the presented data. Missing controls, incompletely described experiments, and statistical irregularities make this manuscript not suitable for publication.

Response: We are grateful to the referee for your valuable comments and constructive suggestions. Now we have seriously considered your comments and performed experiment to improve our findings according to your suggestions.

Major concerns:

1. Sex specific controls

Kaiyu Xu et al. describe a male-specific regulation of gene expression and neuronal plasticity during aging. The authors show several controls for the presented experiments. However, the best and most valid control would be female samples, which should not display the described phenomena. It is essential for the manuscript to add the female control for figures 2 - 7, as well as for s2, s3, s5 - s8.

Response: This point raised by the reviewer is of interest. The requested female control has been added. Our findings suggest that circGRA11 is a male-biased isoform, and the newly *in vivo* knockdown of circGRIA1 further support that circGRIA1 might have little contribution to age-related changes in synaptic plasticity in female macaque, while it will be of interest for us to explore this mystery in next step.

2. Nuclear mechanism of action

The authors base their hypothesis on the nuclear localization of circGRIA1. However, there is no solid evidence for this idea. Controversially, BASEscope staining presented in figures 2a, 2e, 4a, S7a show the localization of the majority of the circRNA outside of the nucleus and the fish experiment presented in S6a needs a Z-stack representation to be fully trusted. The reviewer finds it important to provide nuclear fractionation and demonstrate the purity of the samples via nucleus specific and cytoplasm specific genes.

Response: The point raised by the referee is important. Due to a technical difficulty of microscope for capturing brightfield images, BASEscope ISH of circGRIA1 was not able to process Z-stack deconvolution. Nevertheless, the nuclear fractionation has been

performed for validating the nuclear localization of a majority of circGRIA1(Supplementary Fig 1c).

3. Statistical evaluation

The description of the statistical analysis given by the authors is very often not clear. Moreover, the reviewer is especially concerned about quantification data of cells. In fig c, e, S1b the authors speak of "n=3 repetition of the experiment". Does that mean that the biological n is 1? Further, the authors are referring to "independent experiments". How many individual animals were used per group and per "independent experiment"? It is of very high importance that the authors provide a clear explanation of the biological and technical replicates they used.

In figures 2, 4, s3, s6, s7, the authors counted the intensity of a marker within a cell. From the images presented in the figures, it is hardly possible to find the outline on one given cell. Since the authors used the program Image J for the quantification, the reviewer would ask to see the original images on which the quantification has been performed.

Response: We appreciate the referee's constructive suggestion. All of data have been re-analyzed and labeled clear. Notably, due to the limitation of in vitro culture system, the mechanism of circGRIA1 function in the biology of brain aging has been further performed in vivo. We therefore replaced some of them with new data.

Minor concerns:

4. In f3a, the authors say "either 20-years old male macaque CA1 or fetal macaque hippocampal neurons.". So, what is it?

Response: Thank for referee's careful reviewing. The mistake has been corrected. It should be described as "Total RNA was extracted from postmortem frozen hippocampal tissues of male macaque".

5. In f3d and e the authors compare two populations (10Y and 20Y) to one input. What is the input and how is this comparison valid?

Response: Thank for referee's careful reviewing. Due to quite a low circGRIA1 expression in the brain of 10Y macaque, in this assay, RNA extracts from 20 years was loading input. The label has been corrected.

6. Figures 1f and 3f are sequencing data of several experiments. Why did the authors not present a sequence alignment of the experiments instead of the pure sequence?

Response: We appreciate the referee's good suggestion. The requested sequence alignments have been added.

7. In figures 4a, b, d, and e, the authors are using scrambled siRNA and siRNA-control. What is the difference?

Response: Thank for referee's careful reviewing. The scrambled siRNA from commercial control is commonly used in all siRNA assays, but each of siRNA-controls is specifically designed from the circGRIA1 mismatched pairs.

The information of siRNA-control is described as below (Supplementary Table 4):

Supplementary Table 4. Sequences of siRNAs and BASEScope probes targeting macaque circGRIA1

circRNA	Sequences of siRNAs and BASEScope assay probes	
Gria1	junction site (red)	5'-caugauggcauccgaaagg ^{gg} gcuucauggacauugac-3'
	siRNA-control 1	3'- gaaagccgaaguaccugua -5'
	siRNA 1	3'- cuuuccgaaguaccugua -5'
	siRNA-control 2	3'- uaccguaggcuuucggcuu -5'
	siRNA 2	3'- uaccguaggcuuuc ^{cc} gaa -5'
	BaseScope™ Target Probe	5'-tgattgaaatgaaacatgatggcatccgaaagggttcattg-3' (BA-Mmu-GRIA1-circRNA-Junc.; 1zz targeting 186-227 of the provided sequence >6:150023918 150035468_+)

8. In f4a when OE, siRNA-control, and circGRIA1 siRNA demonstrate a much lower amount of nuclei. How do the authors explain these observations? The amount of red staining is reduced in siRNA-control, how can this be explained? Also, the quality of all the *Gria1* fluorescence images is very bad. The reviewer strongly suggests to use more reliable images with a neuronal counterstaining and clearly visible DAPI.

Response: Thank for referee's careful reviewing. We have replaced it with the new one.

9. In f4b and c the authors performed a t-test on three groups. How is this possible?

Response: Thank for referee's careful reviewing. *p* values were calculated using two-tailed Student's *t* test or one-way ANOVA with Sidak's correction or two-way ANOVA with Turkey's correction for multiple comparisons.

10. In f4d, the loading control for DIV28 siRNA-control and siRNA is strongly overlapping. How did the authors do quantification on such a blot? Also, since this blot has been repeated several times (as indicated in f4e), it is desirable to see all the blots that have been used for the quantification.

Response: Thank for referee's careful reviewing. The mistake of GluR2 band has been replaced with the correct one. All blots from three independent experiment have been listed as below.

11. f5c,d and S8c, d, demonstrate an increase of synapsin I. To validate the data, the authors have to present a western blot and its quantification.

Response: Thank for referee's careful reviewing. The requested data has been added or replaced with the new one.

12. The viral overexpression of circGRIA1 has to be validated by northern blot.

Response: The requested data has been added (The 3rd panel, Supplementary Fig 5a, and Supplementary Fig 6c, d).

13. s3f shows the quantification of the western blot from s3e. Please provide all and the full western blots used for the quantification.

Response: Thank for referee's careful reviewing. All blots from three independent experiments have been listed as below.

14. In s5a and b the authors compare two populations (DIV14 and DIV28) to one input. What is the input and how is this comparison valid?

Response: Thank for referee's careful reviewing. Due to quite a low circGRIA1 expression at DIV14, in this assay, RNA extracts from DIV28 was loading input. The label has been corrected.

15. In s6a and b, the authors used FISH to detect the parental locus of circGRIA1. Since the locus is present in every cell, the authors have to explain why the signal is not clearly visible in every cell.

Response: Thank for referee's careful reviewing. With enhanced contrast in images, the signal of genomic loci (Green dots) is clearly present in every cell.

16. In s6a, the authors describe an experiment with Gapdh. However, Gapdh is not mentioned in the figure at all.

Response: Thank for referee's careful reviewing. *Gapdh* has been labelled now.

REVIEWERS' COMMENTS:

Reviewer #1 (Remarks to the Author):

The authors made a commendable effort by showing that knocking down cirGRIA rescues alterations of synaptic proteins in the aged macaque. Well done
Alfredo Kirkwood

Reviewer #2 (Remarks to the Author):

The manuscript is much improved. Thus I'd like to recommend it to be published.

REVIEWERS' COMMENTS:

Reviewer #1 (Remarks to the Author):

The authors made a commendable effort by showing that knocking down cirGRIA rescues alterations of synaptic proteins in the aged macaque. Well done.

Response: Thank you.

Reviewer #2 (Remarks to the Author):

The manuscript is much improved. Thus I'd like to recommend it to be published.

Response: Thank you.